# PERDUCER: PERSONALIZATION INDUCER FOR TEXT SUMMARIZERS VIA USER PREFERENCE PREDICTION

## ABSTRACT

Document summarization is useful for quick selection and consumption of *highly subjective* content of interest. Identifying *salient* information in a given document, especially one covering multiple aspects, is non-trivial, which further calls for personalized summarization. Modern Large Language Models (LLMs) have shown promising results for in-context-learning-based summarization. However, earlier works have demonstrated their incapability to handle dynamically evolving user-preference histories (in contrast to conventional modeling of static personas). To address this, we propose `PerDucer`, a *summarizer model agnostic personalization booster* that predicts the user's next interaction and thereby generates personalized key-phrases from a given query document. These key-phrases serve as lightweight cues that guide *frozen* summarization models, both small and large. Experiments on the PENS and OpenAI-Reddit datasets reveal that four `PerDucer`-boosted SOTA LLMs outperform their best-performing history-prompt baselines with an average gain of $0.47 \uparrow$ across PSE variants. Two boosted SLMs achieve comparable gains with best (SmolLM2-1.7B) $98.6\%$ of DeepSeek-14B (best LLM) performance.

## 1 INTRODUCTION

In an era of information deluge, modern summarizers help readers assimilate updates rapidly. *Personalized* summarization tailors these updates to the reader's *subjective* interests, a requirement that becomes critical for multi-aspect documents, which must serve diverging foci simultaneously (Dasgupta et al., 2024). Existing studies typically ground personalization in *static* persona attributes—address, gender, nationality, broad topical interests (Dou et al., 2021; He et al., 2022; Li et al., 2023). However, empirical evidence from MS/CAS PENS reveals that user preferences evolve at fine-grained sub-topic levels (Ao et al., 2021). Such long, complex temporal contexts challenge even large language models (LLMs), which otherwise outperform specialized systems on many tasks (Liu et al., 2024; Gao et al., 2024). Indeed, Patel et al. (2024) demonstrated that SOTA LLMs struggle when complex reading histories are injected as prompts in an in-context-learning (ICL) setting; richer reader information, at fixed prompt length, paradoxically degrades performance.

In this paper, we reformulate the history-injected prompt-based approach as ***personalized keyphrase-guided summarization***. We propose `PerDucer` – a Personalization Inducer that serves as a model-agnostic booster to summarizers by providing reader-history-specific keyphrases as cues. `PerDucer` generates ranked personalized keyphrases that summarize the query document in light of the user's evolving reading behavior. Its encoder embeds the reading history as a *temporal user-interaction trajectory*, where nodes are documents and summaries (both model-generated and gold) and edges are transition actions (*click*, *skip*, *read summary*). From this trajectory, the next behavior embedding is predicted, incorporating the query document and its latent personalized summary. The decoder then maps this embedding to a ranked keyphrase list, which is injected into simplified prompts for LLMs within ICL or appended to the query document to induce personalization in otherwise frozen "vanilla" summarizers (Figure 1).

We pose three questions on `PerDucer`'s ability to boost personalization: **RQ-1** *can it improve SOTA LLMs*? **RQ-2** *can it raise SOTA small language models (SLMs) to LLM-like performance*? **RQ-3** *can it push vanilla summarizers past SOTA specialised personalized systems*? For training and evaluation, we use the real-world PENS dataset (Ao et al., 2021) and the synthetic deriva-

tion of the multi-domain OpenAI-Reddit dataset. As of now, PENS is the only available dataset with real-user time-stamped histories. Personalisation is assessed by the three PerSEval variants, PSE-JSD/SU4/METEOR, which align well with human judgement (Dasgupta et al., 2024). For RQ-1: Four frozen LLMs, Mistral-7B (Jiang et al., 2023), Zephyr-7B-$\beta$ (Tunstall et al., 2023), DeepSeek-R1-14B (DeepSeek-AI et al., 2025), and Llama2-Chat-13B (Touvron et al., 2023), gain on average 0.45/0.44/0.53↑ (PSE-JSD/SU4/METEOR) when induced by `PerDucer`. For RQ-2: Two SLMs – SmolLM2-1.7B-Instruct (Allal et al., 2025) and Qwen2.5-0.5B-Instruct (Qwen et al., 2025) – approach LLM scores; SmolLM2 surpasses all LLMs except DeepSeek. For RQ-3: Injecting `PerDucer` into BigBird-Pegasus (Zaheer et al., 2020) and SimCLS (Liu & Liu, 2021) elevates them above the best specialised baseline, GTP (Song et al., 2023); the top configuration (BigBird-Pegasus + `PerDucer`) achieves **0.20/0.11/0.13** ↑. The results confirm that reframing the problem as personalised key-phrase-guided summarisation is highly effective.

## 2 BACKGROUND

**Personalized Summarization.** Personalized summarization aligns outputs with user-specific expectations inferred from temporal behaviors (click, skip, summarize). Traditional accuracy metrics fails to capture this personalization. EGISES (Vansh et al., 2023) addresses this by measuring divergence between expected (gold) and model summaries but ignores model-accuracy gaps. PerSEval, proposed by Dasgupta et al. (2024) refines EGISES by penalizing accuracy drops and is the most stable personalization metric; we therefore use it to evaluate `PerDucer`.

**Training/Evaluation Datasets.** Personalized summarization needs datasets with (i) temporally ordered user interactions, (ii) user-specific gold summaries for shared content, and (iii) diverse, shifting topics/subtopics. CNN/DM (Hermann et al., 2015) and MultiNews (Fabbri et al., 2019) lack user-specific references; OpenAI-Reddit (Völske et al., 2017) lacks temporal interaction sequences. Only PENS (Ao et al., 2021) and PersonalSum (Zhang et al., 2024) meet all criteria. We use PENS, and utilize OpenAI-Reddit with synthetic temporal orders. PENS provides clicks/skips and summaries per user, with averages of 13.6 topics, and a topic-change rate of 0.77, making it a standard benchmark (Ao et al., 2021; Song et al., 2023; Lian et al., 2025).

**Personalized Guided Summarization.** Most personalized-summarization studies assume a *static* user persona. Dou et al. (2021) introduced *GSUM*, which injects user-provided keyphrases restricted to the query document, thus ignoring evolving preferences. *CTRLSum*, *TMWIN*, and *Tri-Agent* similarly rely on static control signals or fixed edit preferences (He et al., 2022; Kirstein et al., 2024; Xiao et al., 2024). PENS augments summarization with external user encoders (NRMS, NAML, EBNR) that capture trajectories but not temporal trends, and remain tied to pointer-generator injections Wu et al. (2019b;a); Okura et al. (2017); Ao et al. (2021). The *GTP* framework (Song et al., 2023) derives latent editing controls from trajectories but its TrRMIo encoder omits short–long term distinctions, unlike `PerDucer`. No prior work differentiates user actions (*click, skip, read-summary*). Signature-Phrase (Cai et al., 2023) reduces trajectories to keyphrases, but temporal dependencies and full interaction patterns remain unmodeled.

## 3 PERSONALIZED SUMMARIZATION: FORMULATION

A key distinction in personalized summarization is between a *static user persona* and a *dynamic user-preference history*. Static persona, such as nationality, address, or broad interests in genres and food, tends to remain relatively unchanged over time. On the other hand, preference histories are highly *dynamic*, since the interaction (or reading behavior) is a temporal sequence, spanning across multiple topics and discourses. Static personal fails to capture the fine-grained variations observed in real-world datasets like PENS (Section 2). To address this, we introduce the *User-Interaction Graph (UIG)*, a data model designed to represent evolving behavior trajectories.

### 3.1 PREFERENCE DATA AS USER–INTERACTION GRAPH (UIG)

We represent user histories as a **User–Interaction Graph** (UIG), a directed acyclic graph $G = \langle N, E \rangle$ where the node set $N$ consists of three disjoint types: (i) **u-nodes** $u^{(t_0)}$ denoting a user at initial timestep $t_0$, (ii) **d-nodes** $d^{(t_p)}$ representing documents interacted at timestep $t_p$, and (iii)

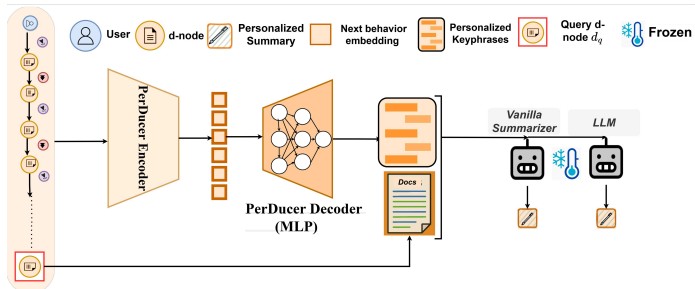

Figure 1: **`PerDucer` Pipeline**: `PerDucer`-**Encoder** predicts the next behavior embedding, which is then fed into a **Key Phrase Extractor** (MLP-based **Decoder**); the extracted top-k key-phrases are used as cues injected into (frozen) summarizers.

**s-nodes** $s_j^{(t_q)}$ representing user-specific summaries requested or generated at time $t_q$ for a document viewed at $t_{q-1}$. The edge set $E$ encodes user actions: $a_d^{(t_p)} \in \{click, skip, summarize\}$ on documents, and $a_s^{(t_q)}$ as the follow-up *summGen* action connecting a document $d^{(t_q-1)}$

**Trajectory**: Given a UIG, the dynamic user preference history (termed *trajectory*) of $u_j$ is a sequence of interactions, denoted $\tau^{u_j}$, starting at $t_0$ and ending at a d-node or s-node at $t_{l-1}$, where $l$ is the trajectory length. Hence, a UIG is a pool of trajectories $\mathcal{T}$ with train-data split denoted as $\mathcal{T}_{\text{train}}$ and test-data split denoted as $\mathcal{T}_{\text{test}}$.

**Behavior Triple**: Given a trajectory $\tau^{u_j}$, a behavior triple at time-step $t$ (denoted $b_{u_j}^{(t_i)}$) is $< hd^{(t_{i-1})}, a^{(t_i)}, tl^{(t_i)} >$ where $hd^{(t_{i-1})}$ denotes *head-node* at time-step $t_{i-1}$, $tl^{(t_i)}$ denotes the *tail-node* at time-step $t_i$, and $a^{(t_i)}$ denotes the user transition *action-relation edge* from $hd$-node to $tl$-node. Note that any $(hd^{(t_{i-1})}, tl^{(t_i)})$ node pair can be either a $(d-d)$, $(d-s)$, or $(s-d)$ node-pair. A UIG can hence be seen as a dynamic temporal knowledge graph (TKG) of user behavior.

**Challenges with LLM-based personalization.** Providing the entire trajectory $\tau^{u_j}$ along with a query document to an LLM for in-context personalization, termed as *In-Context-Personalization-Learning* (ICPL) (Patel et al., 2024), this approach suffers from several limitations. LLMs have a bounded context window and their performance degrades as input length increases, with a well-documented *lost-in-the-middle* effect where information in the middle of long prompts is under-utilized (Chen et al., 2025; Liu et al., 2024; Gao et al., 2024). Empirical studies show that injecting detailed user histories often *reduces* personalization quality, as richer prompts can distract the model and dilute salient cues, illustrating the ICOPERNICUS *Paradox of Less is More* (Patel et al., 2024).

**Problem Formulation.** We therefore reformulate the task into three stages: **Task 1** - predict the next behavior triple $b_{(q,u_j)}$ from $\tau^{u_j}$; **Task 2** - extract personalized key-phrases (top-$k$) from $b_{(q,u_j)}$; and finally, a much simpler **Task 3** - guide the *frozen* summarizer by injecting these key phrases as cues into vanilla models or as prompt context for LLMs.

**Hierarchical Abstraction of UIG** Although TKG-based UIGs are expressive, sequential-recommendation research shows that *hierarchical* abstractions of base actions markedly improve accuracy on very long histories. Layered time-scale graphs that distinguish short- and long-term dependencies (Xia et al., 2022; Ou et al., 2025) and factor-node abstractions of base actions (Xue et al., 2022; Zhang et al., 2022) compress distant influences into compact higher-level states, enabling efficient attention (Ma et al., 2019). Motivated by these findings, we introduce a bi-level UIG: each behaviour triple $b_{u_j}^{(t_i)}$ becomes a **b-node** in a higher-level trajectory, the **b-tier** $\tau_b^{u_j 1}$. Hence, $\tau^{u_j} b$ is the sequence $\langle b^{(t_i)} u_j \rangle$ linked by ***nextBehavior*** edges, and Task 1 is formulated over this structure.

In this work, we construct the UIG u/b-tier from two different sources – (i) PENS forming the train trajectory pool $\mathcal{T}_{\text{train}}^{\text{PENS}}$ and the test $\mathcal{T}_{\text{test}}^{\text{PENS}}$, and (ii) OpenAI-Reddit (Völske et al., 2017) forming $\mathcal{T}_{\text{train}}^{\text{OAI}}$ and $\mathcal{T}_{\text{test}}^{\text{OAI}}$. UIG construction methodology (and algorithm) has been detailed in Appendix B.3.

---

[1]The original sequence is termed the **u-tier**; Detailed notation list: Table 6.

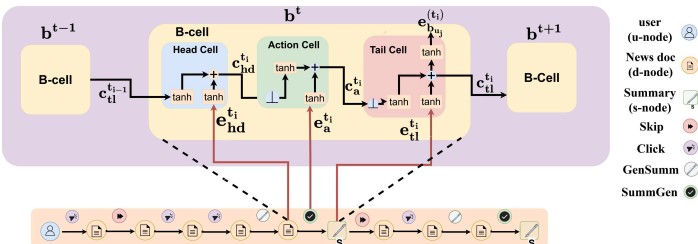

Figure 2: **PerDucer-Encoder**: A **b-node** (behavior triplet) is represented as a **b-cell** containing **head-cell**, **action-cell**, and **tail-cell**. b-cell generates the b-node embedding $e_{b_u}^{(t_i)}$ at timestep $t_i$ using the head-node embedding $e_{hd}^{(t_i)}$ fused to the action-embedding $e_a^{(t_i)}$ via a **projection**, and that then injected into the tail-node embedding $e_{tl}^{(t_i)}$ via another projection.

## 4    PERDUCER: PERSONALIZATION INDUCER FOR SUMMARIZERS

PerDucer is a *personalized keyphrase extractor* that operates in two stages: **Task-1** predicts the next b-node embedding via the encoder, and **Task-2** extracts user-expected keyphrases from this embedding via the decoder. Since such keyphrases can not be directly evaluated, they are passed to an external LM (**Task-3**) with the query document to generate and assess personalized summaries. Thus, instead of end-to-end fine-tuning of LLMs/SLMs, PerDucer focuses on producing high-quality personalized keyphrases for downstream use. Hence, PerDucer acts as a booster by guiding summarizers with personalized key-phrases as cues.

### 4.1    TASK-1: NEXT b-NODE PREDICTION(PERDUCER ENCODER)

**Initialization of u-Tier.** To enable Task 1 at the b-tier, we initialize the u-tier trajectory $\tau^{u_j}$ by embedding each document ($d$) and summary ($s$) node with PromptRank KPE (220M, 768-d) (Kong et al., 2023). For each behavior triple $b_{u_j}^{(t_i)}$, the $hd$ and $tl$ nodes are seeded as $\mathbf{e}_{hd}^{(t_{i-1})}$ and $\mathbf{e}_{tl}^{(t_i)}$. KPE seeding aligns with central themes for keyphrase extraction and outperforms SBERT (Appendix E.2). The initial u-node $\mathbf{e}_{u_j}^{(t_0)}$ uses the title embedding of the first $d$-node to mitigate cold start. Action-transition edges use a 4-d one-hot vector: *click*, *skip*, *summarize*, *summGen*.

**b-Tier Encoder.** PerDucer uses an RNN-style stack of **b-cells** for $\tau_b^{u_j}$. At step $t_i$, a b-cell emits $\mathbf{e}_{b_{u_j}}^{(t_i)}$ and has three sequential components:

(i) the **head-cell**,where the prior tail content $\mathbf{c}_{tl}^{(t_{i-1})}$ with the $hd$-node to get **head-cell content** $\mathbf{c}_{hd}^{(t_i)}$ as follows: $\mathbf{c}_{hd}^{(t_i)} = \tanh\left(W_h \cdot \mathbf{c}_{tl}^{(t_{i-1})} + \mathbf{b}_h\right) + \tanh\left(W_{hd} \cdot \mathbf{e}_{hd}^{(t_i)} + \mathbf{b}_{hd}\right)$

(ii) the **action-cell**, representing one of the four possible transition actions, projects $\mathbf{c}_{hd}^{(t_i)}$ onto the action hyperplane, inspired by Wang et al. (2014) to generate $\mathbf{c}_a^{(t_i)}$:

$$\mathbf{c}_a^{(t_i)} = \tanh\left(W_h \cdot \text{proj}_{\mathbf{e}'^{(t_i)}_a} \mathbf{c}_{hd}^{(t_i)} + \mathbf{b}_{hd\perp a}\right) + \mathbf{e}'^{(t_i)}_a; \quad \mathbf{e}'^{(t_i)}_a = \tanh\left(W_a \cdot \mathbf{e}_a^{(t_i)} + \mathbf{b}_a\right) \quad (1)$$

(iii) the **tail-cell** finally fuses $\mathbf{c}_a^{(t_i)}$ with the $tl$-node embedding $\mathbf{e}_{tl}^{(t_i)}$ by projecting back $\mathbf{c}_a^{(t_i)}$ onto the node-hyperplane to form the **tail-cell content** $\mathbf{c}_{tl}^{(t_i)}$ as:

$$\mathbf{c}_{tl}^{(t_i)} = \tanh\left(W_h \cdot \text{proj}_{\mathbf{e}'^{(t_i)}_{tl}} \mathbf{c}_a^{(t_i)} + \mathbf{b}_{a\perp tl}\right) + \mathbf{e}'^{(t_i)}_a; \quad \mathbf{e}'^{(t_i)}_{tl} = \tanh\left(W_{tl} \cdot \mathbf{e}_{tl}^{(t_i)} + \mathbf{b}_{tl}\right) \quad (2)$$

The tail-cell content $\mathbf{c}_{tl}^{(t_i)}$ represents the content of the b-cell flowing onto the next b-cell. *The last b-cell content embedding represents* $\tau^{(u_j)}$. The b-node embedding is $\mathbf{e}_{b_{u_j}}^{(t_i)} = \tanh\left(W_b \cdot \mathbf{c}_{tl}^{(t_i)} + \mathbf{b}_b\right)$. While $\mathbf{e}_{b_{u_j}}$ captures fine-grained behavior semantics at each step, it remains a *local representation* sensitive to the current behavior and near-past historical span.

**History Aware Encoding via Decay-EMA.** Building on MEGA's damped-EMA (Ma et al., 2023), we introduce a *content-aware* Decay-EMA (D-EMA) that tracks slow interest drift by blending the current behavior with a smoothed history to form a *cumulative "snapshot"* representation $\mathbf{e}_{b_{u_j}^{\text{D-EMA}}}^{(t_{1:i})}$ as:

$$
\begin{aligned}
\mathbf{e}_{b_{u_j}^{\text{D-EMA}}}^{(t_{1:i})} &= \alpha^{(t_i)} \odot \mathbf{e}_{b_{u_j}}^{(t_i)} + (1 - \alpha^{(t_i)} \odot \delta^{(t_i)}) \odot \mathbf{e}_{b_{u_j}^{\text{D-EMA}}}^{(t_{1:i-1})}; \\
\alpha^{(t_i)} &= \tanh\left(W_\alpha \cdot [\mathbf{e}_{b_{u_j}^{\text{D-EMA}}}^{(t_{i-1})}; \mathbf{e}_{b_{u_j}^{\text{D-EMA}}}^{(t_i)}] + \mathbf{b}_\alpha\right); \quad \delta^{(t_i)} = \tanh\left(W_\delta \cdot [\mathbf{e}_{b_{u_j}^{\text{D-EMA}}}^{(t_{i-1})}; \mathbf{e}_{b_{u_j}^{\text{D-EMA}}}^{(t_i)}] + \mathbf{b}_\delta\right)
\end{aligned}
\tag{3}
$$

Here, $\alpha^{(t_i)}$ is a learnable content-aware decay, and $\delta^{(t_i)}$ is a content-aware damping gate. They enable adaptive control over how much recent history influences the state at $t_i$. To illustrate, consider Alice's trajectory: at $t_1$ she **clicks** on *global-markets*, at $t_2$ **skips** *celebrity-gossip*, and at $t_3$ **clicks** on *AI-policy*. At $t_4$ she requests a **summary** of that piece, receives it at $t_5$, and at $t_6$ clicks into a related *semiconductors* article. By $t_7$ she **skips** a *sports-roundup*, and at $t_8$–$t_{10}$ returns to *AI-policy* with further clicks and summaries. D-EMA blends these steps into cumulative snapshots where repeated interest in *AI-policy* is reinforced, while distractions like *celebrity-gossip* or *sports-roundup* are down-weighted. However, sequential blending still fails to capture *non-local dependencies*. In Alice's case, her renewed attention to *AI-policy* at $t_8$ is semantically tied to her earlier click at $t_3$, despite intervening detours.

**Contextualizing D-EMA with Self-Attention.** We address the above by enriching D-EMA with forward-masked self-attention (FM-Attn) to model long-range dependencies among cumulative snapshots. Given the residual transform $\mathbf{e}'^{(t_{1:i})}_{b_{u_j}^{\text{D-EMA}}} = W_{\text{D-EMA}} \mathbf{e}_{b_{u_j}^{\text{D-EMA}}}^{(t_{1:i})} + \mathbf{b}_{\text{D-EMA}}$, the contextualized state is:

$$
\mathbf{e}_{b_{u_j}^{\text{c-EMA}}}^{(t_{1:i})} = \phi_{\text{SiLU}}\left(W_{\text{c-EMA}} \cdot \left(\phi_{\text{SiLU}}\left(\mathbf{e}'^{(t_{1:i})}_{b_{u_j}^{\text{D-EMA}}}\right) + \mathbf{f}^{(t_i)} \odot \mathbf{FM\text{-}Attn}\left(\mathbf{e}_{b_{u_j}^{\text{D-EMA}}}^{(t_{1:i})}\right)\right) + \mathbf{b}_{\text{c-EMA}}\right)
\tag{4}
$$

forget gate at $t_i$: $\mathbf{f}^{(t_i)} = \phi_{\text{SiLU}}\left(W_f \cdot \mathbf{e}'^{(t_{1:i})}_{b_{u_j}^{\text{D-EMA}}} + \mathbf{b}_f\right); \mathbf{e}'^{(t_{1:i})}_{b_{u_j}^{\text{D-EMA}}} = W_{\text{D-EMA}} \cdot \mathbf{e}_{b_{u_j}^{\text{D-EMA}}}^{(t_{1:i})} + \mathbf{b}_{\text{D-EMA}}; \mathbf{e}'^{(t_{1:i})}_{b_{u_j}^{\text{D-EMA}}}$

For Alice, this means that her renewed interest in *AI-policy* at $t_8$–$t_{10}$ can explicitly attend back to the earlier interaction at $t_3$, rather than relying only on the sequentially decayed trace. FM-Attn therefore captures her *cyclical preference* – a hallmark of real-world behavior where themes re-emerge after gaps. Finally, we add a calibrated residual (using the input gate $\mathbf{i}$) to recount the current time-step b-node information, and generate the **content-aware MEGA (c-MEGA)** representation of $b_{u_j}^{t_i}$ as:

$$
\mathbf{e}_{b_{u_j}^{\text{c-MEGA}}}^{(t_i)} = \mathbf{i}^{(t_i)} \odot \mathbf{e}_{b_{u_j}^{\text{c-EMA}}}^{(t_{1:i})} + (1 - \mathbf{i}) \odot \mathbf{e}_{b_{u_j}}^{(t_i)}; \quad \mathbf{i}^{(t_i)} = \sigma\left(W_i \cdot \mathbf{e}'^{(t_{1:i})}_{b_{u_j}^{\text{D-EMA}}} + \mathbf{b}_i\right)
\tag{5}
$$

**Predicting Next b-Node.** Given the final contextualized b-node embedding $\mathbf{e}_{b_{u_j}^{\text{c-MEGA}}}^{(t_l)}$ (where $l$ is the length of $\tau^{u_j}$), we apply a prediction head to obtain the **query** b-node at $t_{l+1}$ as:

$$
\mathbf{e}_{b_{u_j}^{\text{q}}}^{(t_{l+1})} = W_{\text{pred}} \mathbf{e}_{b_{u_j}^{\text{c-MEGA}}}^{(t_l)} + \mathbf{b}_{\text{pred}}.
$$

For Alice, this corresponds to predicting that, after her latest sequence ending at $t_{10}$ (summarizing *AI-policy*), her next likely behavior at $t_{11}$ will again involve clicking on a related *AI-policy* document – say, a committee report – since the contextualized state has reinforced this thematic preference through both local evidence and long-range attention. The action-cell content of $\mathbf{e}_{b_{u_j}^{\text{c-MEGA}}}^{(t_l)}$ includes the embedding of the *genSumm* action on the query document $d_q^{(t_l)}$, which is integrated within the tail-cell content (see Figure. 3; Details in Appendix A.3).

### 4.2 Task-2: Personalized Key Phrase Extraction (PerDucer Decoder)

**MLP Decoder for Key-Phrases.** In the final `PerDucer` step, the predicted query b-node $\mathbf{e}_{b_{u_j}^{\text{q}}}^{(t_{l+1})}$ is mapped by an MLP to a distribution over a KPE-derived key-phrase vocabulary.[2] The decision

---

[2] YAKE Ricardo Campos (2020) is applied on PENS train to build the vocabulary; size: 2680K.

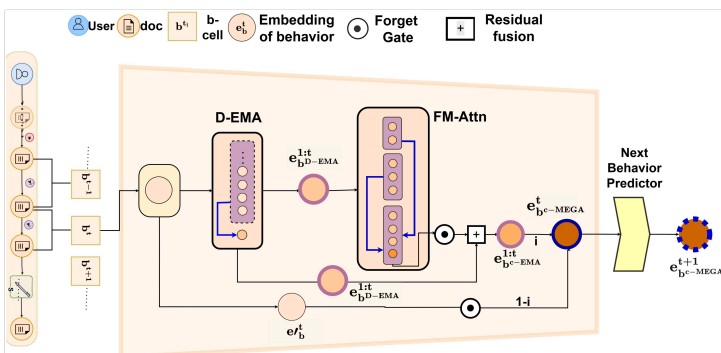

Figure 3: **PerDucer Encoder**: b-node progressive enrichment via D-EMA, c-EMA, and c-MEGA. For b-cell architecture see Figure 2.

head outputs $\hat{\mathcal{P}}_{KP}$, from which we select the top-$k$ phrases:

$$\hat{\mathcal{P}}_{KP} = \textbf{SoftMax}\left(\text{MLP}(\mathbf{e}^{(t_{l+1})}_{b^q_{u_j}})\right); \quad \{kp\}_k = argsort_k(\hat{\mathcal{P}}_{KP}) \tag{6}$$

Compilations of all `PerDucer` notations, parameters, and hyperparameters are in Tables 6 and 10.

### 4.3 TASK 3: GUIDED PERSONALIZED SUMMARIZATION VIA KEY-PHRASE INJECTION

To assess and guide the frozen summarizers, we incorporate task 3 – to feed the extracted top-$k$ key-phrases into a summarizer for boosting personalization.

**Vanilla Summarizers.** Following Vansh et al. (2023), we score each sentence in $d_q$ by key-phrase frequency, select the top-$m$ *theme sentences*, and prepend: [Doc. Body: $\cdots$ ; Theme Sentences: $\cdots$] before encoding $d_q$.

**Large (& Small) Language Models.** For LLMs/SLMs, we provide key-phrases directly in the prompt: [Task: $\cdots$ ; Document: $\cdots$ ; Key-Phrases: $\cdots$ ; Conditions: $\cdots$] (templates: Appx. F).

## 5 EVALUATION

### 5.1 TRAINING SETUP

**Training Data.** We build UIGs from PENS ($\mathcal{T}^{\text{PENS-D}}$) and OpenAI-Reddit ($\mathcal{T}^{\text{OAI}}$) (Appendix B.3). From these, we sample 150K PENS trajectories ($\overline{|d|} = 123$, $\overline{|s|} = 15$) and 45K OAI trajectories ($\overline{|d|} = 37$, $\overline{|s|} = 12$) for training.[3] Each train instance slices a trajectory before a $(d-s)$ pair to form user history $\tau^{u_j}_{\text{h}}$, query $d_q$, and target summary $s^*_q$.

**Test Data.** For PENS test $\mathcal{T}^{\text{PENS-D}}_{\text{test}}$, we merge clicked docs (stage-1) with $(d-s)$ pairs (stage-2), then create 150 test trajectories, with 150 trajectories per 103 users ($\approx 15k$ test rows) by sliding a cut after the first 50 pairs: $\tau^{u_j}_{\text{h}}$ ends at pair $t$, $d_q$ is the next $d$, and $s^*_q$ its $s$ (Fig. 4). For openAI test $\mathcal{T}^{\text{OAI}}_{\text{test}}$, we sample 10K trajectories and slice before each $(d-s)$ pair to obtain $(\tau^{u_j}_{\text{h}}, d_q, s^*_q)$.

**PerDucer Training.** `PerDucer` is trained with losses defined at two levels – the Decoder loss (i.e., the KPE Loss ($\mathcal{L}_{\text{KPE}}$)) and the Encoder Loss ($\mathcal{L}_{\text{ENC}}$). We first extract all the top-$k$ key-phrases ($\{kp_i\}_{1:k}$) in the target summary $s^*_q$. We create a **target multi-hot label vector** $\mathbf{1}_{(k \times 1)}$, where each component represents probability of 1 for each extracted target key-phrase from the ground-truth s-node (by SpaCy v3). We then apply Mean NLL Loss as: $\mathcal{L}_{\text{KPE}} = -\frac{1}{k} \cdot \sum_{i=1}^{k} \log \hat{p}(kp_i)$;    ideally, $\hat{p}(kp_i) = 1$. $\mathcal{L}_{\text{KPE}}$ is backpropagated to the Encoder and gets added up

---

[3]**Sizes:** $\mathcal{T}^{\text{PENS-D}}$: 360K, $\mathcal{T}^{\text{OAI}}$: 45K; **Max steps:** PENS train 200, OAI train 50.

Table 1: **RQ-1/2: `PerDucer`-boost consistency across LLMs/SLMs for all architectural progressions on PENS dataset; OpenAI-Reddit results in Table 5** *Obs.-1: c-MEGA outperforms all versions* of `PerDucer`, *highlighting the need for Residual Fusion of D-EMA with D-EMA+FM-Attn*; *Obs.-2: DeepSeek leads, but SLMs are competitive on narrower tasks*; Stat. sig. $p \leq 0.05$.

| LLM/SLM | 2-shot | | | B-tier Vanilla | | | D-EMA | | | D-EMA+FM-Attn | | | C-MEGA | | |
|---|---|---|---|---|---|---|---|---|---|---|---|---|---|---|---|
| | JSD | SU4 | METEOR | JSD | SU4 | METEOR | JSD | SU4 | METEOR | JSD | SU4 | METEOR | JSD | SU4 | METEOR |
| Mistral-7B | 0.23 | 0.09 | 0.08 | 0.48 | 0.27 | 0.31 | 0.59 | 0.35 | 0.42 | 0.57 | 0.38 | 0.44 | 0.67 | 0.52 | 0.6 |
| DeepSeek-R1 | 0.24 | 0.09 | 0.1 | 0.51 | 0.292 | 0.32 | 0.6 | 0.362 | 0.43 | 0.58 | *0.39* | *0.45* | **0.71** | **0.54** | **0.62** |
| Zephyr-7B-$\beta$ | 0.23 | 0.08 | 0.08 | 0.5 | 0.28 | 0.32 | 0.56 | 0.35 | 0.4 | *0.59* | 0.36 | 0.43 | 0.69 | 0.53 | 0.6 |
| LLaMA-13B | 0.22 | 0.07 | 0.08 | 0.43 | 0.26 | 0.3 | 0.48 | 0.36 | 0.41 | 0.5 | 0.37 | 0.43 | 0.68 | 0.53 | 0.61 |
| Qwen2.5-0.5B | NA* | NA* | NA* | 0.34 | 0.23 | 0.26 | 0.55 | 0.32 | 0.39 | 0.52 | 0.33 | 0.38 | 0.65 | 0.46 | 0.58 |
| smolLM2-1.5B | NA* | NA* | NA* | 0.43 | 0.28 | 0.33 | 0.59 | 0.36 | 0.42 | 0.54 | 0.38 | 0.44 | **0.7** | **0.53** | **0.61** |

with the auxiliary $\mathcal{L}_{\text{ENC}}$ as: $\mathcal{L}_{\text{PerDucer}} = \alpha \cdot \mathcal{L}_{\text{KPE}} + (1-\alpha) \cdot \mathcal{L}_{\text{ENC}}$. $\mathcal{L}_{\text{ENC}}$ is the loss defined on the incorrect encoding of the provided user history $\tau_h^{u_j}$ of length $l$. We add a learnable **position extractor** $W_{\text{pos}}$ on each b-node embedding $\mathbf{e}_{b_{u_j}^{\text{c-MEGA}}}^{(t_i)}$ to generate the occurrence probability distribution $\hat{\mathcal{P}}_{\text{pos}}$ of

$b^{(t_i)}$ over all possible steps $i = [1 : l_{\max}]$ as: $\hat{\mathcal{P}}_{\text{pos}} = \textbf{SoftMax}\left(W_{\text{pos}} \cdot \mathbf{e}_{b_{u_j}^{\text{c-MEGA}}}^{(t_i)}\right)$; ideally, $\hat{p}(t_i) = 1$.

Hence, $\mathcal{L}_{\text{ENC}}$ can be defined across $\tau_h^{u_j}$ for each time-step $t_i$ as: $\mathcal{L}_{\text{ENC}} = -\sum_{i=1}^{l} \log \hat{p}(t_i)$. In our train dataset, $l_{\max} = 200$. $W_{\text{pos}}$ explicitly aligns each b-cell embedding to its actual time-step.

## 5.2 BASELINE SUMMARIZATION MODELS

**LLMs-as-summarizers.** For RQ-1, we benchmark four frozen LLMs—Mistral-7B-Instruct (Jiang et al., 2023), DeepSeek-R1-Distill-Qwen-14B (DeepSeek-AI et al., 2025), LLaMA-2-13B-Chat-HF (Touvron et al., 2023), and Zephyr-7B (Tunstall et al., 2023)—using the strongest 0-/2-shot prompts from Patel et al. (2024) and prompt chaining where applicable. Rather than seeking a "best LLM," our aim is to show that `PerDucer` consistently boosts frozen LLMs, **acting as a model-agnostic personalization adapter *without retraining***. Any LM that stands out can be further PEFT-tuned as per case constraints. Hence, we also pose `PerDucer` as an energy/cost-efficient selection method.

**Non-personalized summarizers with cue injection (Oracle).** We also include two generic SOTA summarizers: BigbirdPegasus (Zaheer et al., 2020) and SimCLS (Liu & Liu, 2021) (RQ-1b). Following Vansh et al. (2023), we augment the query with gold cues, effectively giving these models an oracle-style upper bound on personalization.

**Small language models.** For RQ-2, we test frozen SLMs Qwen2.5-0.5B-Instr. (Qwen et al., 2025) and SmolLM2-1.7B-Instr. (Allal et al., 2025). Their limited context windows make them ideal for comparison against boosted LLMs under identical conditions.

**Personalized summarizers.** Finally, for RQ-3 we evaluate three SOTA personalized frameworks: PENS (Ao et al., 2021), GTP (Song et al., 2023), and Signature-Phrase (Cai et al., 2023). PENS uses external user encoders (Transformer-based NAML (Wu et al., 2019a) and NRMS (Wu et al., 2019b), and GRU-based EBNR (Okura et al., 2017)). GTP integrates Transformer-based TrRMIo internally, and since Signature-Phrase models user-specific keyphrases, it is an important baseline. All baselines are fine-tuned end-to-end for two epochs ***under the same training regime*** as **`PerDucer`**.

All baseline details are in Appendix C.

## 5.3 EVALUATION METRICS

To evaluate the efficacy of `PerDucer` w.r.t the boost in the degree-of-personalization, we choose PerSEval (PSE), the only known evaluation metric for personalized summarization proposed by Dasgupta et al. (2024). Also, results therein show that PerSEval explicitly captures accuracy, thereby rendering a separate accuracy leaderboard redundant. PSE-SU4/METEOR/JSD are selected as the three variants due to their high human-judgment correlation and computational efficiency.

**Accuracy Evaluation.** We report standard content-overlap scores (ROUGE-SU4 (Lin, 2004), ROUGE-L (Lin & Och, 2004)) against gold summaries. We complement the intrinsic metrics with human rating judgment. We assess how generated summaries align with what users **prefer**. Us-

Table 2: **Personalized Summarization Performance w.r.t Accuracy & Human-Judgment Ratings:** Avg. interpolated rating on OpenAI (Reddit) dataset; Details in Table 14; Stat. sig. $p \leq 0.05$.

| Category | Model | Rouge-SU4 | Rouge-L | HJ-Interpolated Ratings |
|---|---|---|---|---|
| Best Specialized (Personalized) | PENS-NRMS-T2 | 13.64 | 21.03 | 2 |
| | GTP-TrRMIo | 21.91 | 28.31 | 2 |
| | SP-Individual | 19.54 | 25.18 | 3 |
| Best LLMs (2-shot history) | DeepSeek-14B | 19.57 | 29.72 | 5 |
| | LLaMA-2 | 18.31 | 29.54 | 5 |
| Best in `PerDucer` | `PerDucer`+DeepSeek14B | **65.14** | **67.82** | 7 |
| | `PerDucer`+LLaMA | **63.55** | **67.16** | 7 |

Table 3: **RQ-3: Performance of Vanilla Summarizers and Comparison w.r.t. SOTA specialized models on PENS dataset.** *Observation-1:* `PerDucer`*-guided keyphrases boost them to near parity personalization in terms of Vanilla Models as Upper Bound Oracle; Observation-2: Boosted Vanilla outperforms all baseline SOTA personalized summarizers;* Stat. sig. $p \leq 0.05$.

| Type | Model | PSE-JSD | PSE-SU4 | PSE-METEOR |
|---|---|---|---|---|
| Specialized Models | PENS-NAML-T1 | 0.021 | 0.014 | 0.016 |
| | PENS-EBNR-T1 | 0.015 | 0.010 | 0.011 |
| | PENS-EBNR-T2 | 0.011 | 0.008 | 0.009 |
| | PENS-NRMS-T1 | 0.015 | 0.011 | 0.011 |
| | PENS-NRMS-T2 | 0.008 | 0.007 | 0.007 |
| | GTP | 0.024 | 0.017 | 0.019 |
| | SP-Individual | 0.017 | 0.015 | 0.014 |
| Generic + Title (Oracle) | BigbirdPegasus | *0.253* | *0.143* | *0.168* |
| | SimCLS | 0.157 | 0.032 | 0.016 |
| Generic + `PerDucer` Keyphrase | BigbirdPegasus | **0.228** | **0.136** | **0.154** |
| | SimCLS | 0.104 | 0.026 | 0.014 |

ing the multi-domain non-news OpenAI-Reddit dataset, which contains multiple human-rated summaries of 9 models, we identify the top-rated (i.e., 7) one per user as the *human-preferred reference*. We then measure the SBert-embedding-space RMSD-divergence of the model-generated summaries from the reference and create a ground rating-to-RMSD-range map table, where each rating row has its corresponding average min-max range. Using this table, we interpolate the HJ-rating of our baseline models as in Table 14.

## 6   RESULTS: PERSONALIZATION BOOST CONSISTENCY

### 6.1   RQ-1: PERFORMANCE W.R.T. BOOSTING LLMs (PERSONALIZATION GAIN)

We evaluate how effectively `PerDucer` boosts personalization capabilities of the baseline LLMs (temperature: 0.2 to ensure faithfulness; details: Appendix E.2.) when compared to their 2-prompt-based baseline (Section 5.2) performance (prompt structure comparison details: Appendix F). The default top-$k$ key-phrases extracted are 10. We find a *significant improvement* in personalization performance, with average gains of **0.45/0.44/0.53** ↑ w.r.t PSE-JSD/SU4/METEOR, respectively (Table 1). The results strengthen our claim that *simplifying the personalized summarization task is a more promising direction*. We also observe that `PerDucer`-boosted LLMs beat best LLM baseline (DeepSeek-14B) in both the accuracy metrics with 0.42 and 0.38 boost w.r.t Rouge-SU4 and Rouge-L (see Table 11), and that it achieves 7/7 in terms of human ratings (see Table 14).

**Inducing Personalization in Vanilla Summarizers (Approximating Oracle).** In order to analyze the performance of personalized KPE (task-2), we compare the PSE-scores of the personalized key-phrases injected vanilla summarizers (Section 4.3) with their corresponding oracle version's performance (as described in Section 5.2). We find that `PerDucer` boosts the models close to their best-possible PSE-scores, with the best result (BigBirdPegasus) achieving **90.12/95.1/91.67%** of the oracle-performance w.r.t PSE-JSD/SU4/METEOR (Table 3 for detailed results).

### 6.2   RQ-2: PERFORMANCE W.R.T. BOOSTING SLMs (PERSONALIZATION GAIN)

It has been observed that Small Language Models (SLMs) can approximate the performance of LLMs on specific, simpler tasks (Fu et al., 2024; Xu et al., 2025). Since the personalized summarization task has been reduced to guided summarization, we analyze the SOTA baseline SLMs when boosted with `PerDucer` (Table 1). We find that SmolLM2-1.7B-Instruct slightly outperforms 3 LLMs, except DeepSeek, where it achieves near-parity with a marginal difference of **0.01** w.r.t PSE-

Table 4: **Top-$k$ Key-phrase Ablation:** $k = 10$ consistently outperforms; Stat. sig. $p \leq 0.05$.

| LLMs | 5 Keyphrases | | | 10 Keyphrases | | | 15 Keyphrases | | |
|---|---|---|---|---|---|---|---|---|---|
| | PSE-JSD | PSE-SU4 | PSE-METEOR | PSE-JSD | PSE-SU4 | PSE-METEOR | PSE-JSD | PSE-SU4 | PSE-METEOR |
| Mistral-7B | 0.075 | 0.045 | 0.052 | 0.676 | 0.524 | 0.604 | 0.632 | 0.523 | 0.573 |
| DeepSeek-R1 | 0.077 | 0.048 | 0.055 | **0.710** | **0.543** | **0.627** | 0.682 | 0.540 | 0.611 |
| Zephyr-7B-$\beta$ | 0.066 | 0.044 | 0.051 | 0.695 | 0.530 | 0.607 | 0.673 | 0.503 | 0.587 |
| LLaMA-13B | 0.065 | 0.043 | 0.039 | 0.685 | 0.533 | 0.614 | 0.671 | 0.532 | 0.413 |
| Qwen2.5-0.5B | 0.063 | 0.037 | 0.039 | 0.652 | 0.467 | 0.585 | 0.658 | 0.477 | 0.537 |
| smolLM2-1.5B | 0.068 | 0.047 | 0.054 | 0.700 | 0.536 | 0.615 | 0.628 | 0.515 | 0.586 |

Table 5: **Cross-domain Generalizability:** `PerDucer` trained in OpenAI-Reddit; $p \leq 0.05$.

| Model | w/ history | | | PENS Test | | | OpenAI Test | | |
|---|---|---|---|---|---|---|---|---|---|
| | JSD | SU4 | METEOR | JSD | SU4 | METEOR | JSD | SU4 | METEOR |
| DeepSeek-R1 | 0.243 | 0.095 | 0.109 | **0.517** | 0.374 | 0.437 | **0.632** | **0.473** | **0.524** |
| Zephyr-7B-$\beta$ | 0.214 | 0.087 | 0.104 | 0.485 | 0.352 | 0.373 | 0.624 | 0.471 | 0.518 |
| LLaMA-13B | 0.232 | 0.093 | 0.107 | 0.504 | **0.381** | **0.451** | 0.627 | **0.473** | 0.521 |
| Mistral-7B | 0.226 | 0.088 | 0.103 | 0.487 | 0.362 | 0.418 | 0.612 | 0.452 | 0.504 |
| smolLM2-1.5B | NA* | NA* | NA* | 0.513 | 0.373 | 0.431 | 0.628 | 0.470 | 0.521 |
| Qwen2.5-0.5B | NA* | NA* | NA* | 0.476 | 0.343 | 0.406 | 0.584 | 0.434 | 0.458 |

JSD/SU4/METEOR, and Qwen2.5-0.5B-Instruct trails behind at an average of just **0.06/0.08/0.04** w.r.t PSE-JSD/SU4/METEOR. The results show that `PerDucer` *effectively boosts SLMs to approximate LLMs w.r.t personalized summarization*, given that the SLMs are incapable of exhibiting ICL via prompt-based history injection. This again supports that *reducing the problem to personalized guided summarization is a superior approach* (see Table 14 for qualitative assessment).

**RQ-1/2 Ablation: Effectiveness of c-MEGA-based User Preference Modeling.** In order to understand the effect of the c-MEGA architecture of `PerDucer` Encoder, we ablate on all the four design progressions as described in Section A.3.1– (i) vanilla b-tier (without any EMA modeling or FM-Attn), (ii) b-tier+D-EMA, (iii) b-tier+FM-Attn (i.e., c-EMA), and (iv) c-MEGA. We observe that c-MEGA outperforms all other versions with a margin of **0.128/0.154/0.171**↑ w.r.t PSE-JSD/SU4/METEOR in comparison to the next best c-EMA version (for details, see Table 1).

**RQ-1/2 Ablation: Key-Phrase Count.** We vary extracted key-phrases [5,10,15] (ground-truth avg.: 20.23) and find k=10 performs best, giving gains of **0.32/0.25/0.29**↑ on PSE-JSD/SU4/METEOR (Table 4). We match the generated keyphrases w.r.t. ground truth keyphrase tokens and find a match of $78.76\%$ and $0.8$ Precision/Recall, underscoring the quality of personalized keyphrase generation.

**Cross-domain applicability.** We train `PerDucer` on OpenAI-Reddit $\mathcal{T}_{\text{train}}^{\text{OAI}}$ (29 non-news domains) and test $\mathcal{T}_{\text{test}}^{\text{OAI}}$, where c-MEGA yields strong gains (e.g., Mistral-7B: $0.386/0.364/0.4$ ↑). To validate domain transfer, we further test on real PENS ($\mathcal{T}_{\text{test}}^{\text{PENS-D}}$), still observing notable improvements ($0.26/0.27/0.32$ ↑), confirming `PerDucer`'s generalizability beyond news-centric data (Table 5).

## 6.3 RQ-3: BOOSTED VANILLA SUMMARIZERS W.R.T PERSONALIZED SUMMARIZERS

We study the efficacy of `PerDucer` as a booster by further comparing the personalization induced in the (*frozen*) vanilla summarizers with that of specialized *finetuned* baseline models. We observe that the induced BigBird-Pegasus outperforms the best-performing specialized model (GTP) by a massive margin of **0.196/0.11/0.131** w.r.t PSE-JSD/SU4/METEOR. This further reinforces that *reducing the problem to personalized guided summarization is more effective* than pure self-attention, or RNN-styled user history modeling as adopted by GTP and PENS (see Table 3).

## 7 CONCLUSION

Personalized summarization is challenging due to long, mixed user histories combining positive (*click*, *read*) and negative (*skip*) signals. Current LLMs struggle to encode such structures in-context. In this paper, we propose `PerDucer`, which reframes the task as *personalized key-phrase-guided summarization*: user-behavior encoders predict a latent personalized embedding, decode it into key phrases, and inject them into the summarizer. Experiments show consistent personalization gains (**0.44**↑ on average), especially for LLMs otherwise weak at personalization. Preliminary studies on recommendations beyond summarization are in Table 15. Remaining challenges include cross-domain data scarcity and ensuring safeguards against leakage and opinion manipulation.

## CODE OF ETHICS STATEMENT

This research adheres to the ICLR Code of Ethics[4]. In conducting this work, we: (i) contributed to society and human well-being by advancing methods for trustworthy personalized summarization; (ii) upheld high standards of scientific excellence through transparent reporting, reproducibility, and acknowledgement of prior work; (iii) avoided harm by ensuring that our methods were tested responsibly, with no foreseeable misuse to compromise safety, security, or privacy; (iv) were honest, trustworthy, and transparent in disclosing our methods, limitations, and potential risks; (v) acted fairly and without discrimination, considering inclusivity in data and evaluation; (vi) respected the work and rights of others via proper citation and intellectual property compliance; and (vii) respected privacy by not using personally identifiable or sensitive information in our datasets. (viii) used LLMs (GPT-5) limited to structural changes (paraphrasing and summarization of our own content, which has not been used verbatim in most of the paper), table format corrections, and extensive literature review (using Deep Research). We have not used LLM for any content *generation* purpose.

## REPRODUCIBILITY STATEMENT

We have made significant efforts to ensure the reproducibility of our work. All details of the proposed `PerDucer` framework, including the encoder and decoder variants, training objectives, and evaluation protocols, are provided in Sections 4 and 5. Hyperparameter choices, model configurations, and training details are documented in Appendix D.2 and Table 10. Dataset descriptions, preprocessing steps, and evaluation metrics (PSE-JSD, SU-4, METEOR) are clearly specified in Sections 2 and 5.3, Appendices B and A.2.1. We also provide ablation studies (Tables 4, 5, & 13) to demonstrate robustness to design choices. To facilitate independent verification, we include a zip file of our source code and scripts in the supplementary material, which allows reproduction of all reported experiments.

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

## A   MEASURING DEGREE-OF-PERSONALIZATION

### A.1   MOTIVATION

Vansh et al. (2023) proposed EGISES– a metric to measure the degree of **in**sensitivity-to-subjectivity for relative benchmarking of how much models *lack personalization* (i.e., a lower score is better within the range $[0, 1]$) instead of assigning an absolute goodness score. Based on this notion, they defined (summary-level) "**deviation**" of a model $M_{\boldsymbol{\theta},u}$(later termed as ***Degree-of-Responsiveness*** (DEGRESS) by Dasgupta et al. (2024)) as follows:

**Summary-level DEGRESS.** Given a document $d_i$ and a user-profile $u_{ij}$ (user $j$'s expected summary), the summary-level responsiveness of a personalized model $M_{\boldsymbol{\theta},u}$, (i.e., DEGRESS($s_{u_{ij}}|(d_i, u_{ij})$)), is defined as the *proportional* divergence between model-generated summary $s_{u_{ij}}$ of $d_i$ for $j$-th user from other user-specific summary versions w.r.t a corresponding divergence of $u_{ij}$ from the other user-profiles.

DEGRESS($s_{u_{ij}}|(d_i, u_{ij})$) is formulated as:

$$\text{DEGRESS}(s_{u_{ij}}|(d_i, u_{ij})) = \frac{1}{|\mathbf{U}_{d_i}|} \sum_{k=1}^{|\mathbf{U}_{d_i}|} \frac{min(X_{ijk}, Y_{ijk}) + \epsilon}{max(X_{ijk}, Y_{ijk}) + \epsilon}$$

$$X_{ijk} = \frac{\exp(w(u_{ij}|u_{ik}))}{\sum\limits_{l=1}^{|\mathbf{U}_{d_i}|} \exp(w(u_{ij}|u_{il}))} \cdot \sigma(u_{ij}, u_{ik}); \ Y_{ijk} = \frac{\exp(w(s_{u_{ij}}|s_{u_{ik}}))}{\sum\limits_{l=1}^{|\mathbf{U}_{d_i}|} \exp(w(s_{u_{ij}}|s_{u_{il}}))} \cdot \sigma(s_{u_{ij}}, s_{u_{ik}}) \quad (7)$$

$$w(u_{ij}|u_{ik}) = \frac{\sigma(u_{ij}, u_{ik})}{\sigma(u_{ij}, d_i)}; \ \ w(s_{u_{ij}}|s_{u_{ik}}) = \frac{\sigma(s_{u_{ij}}, s_{u_{ik}})}{\sigma(s_{u_{ij}}, d_i)}$$

Here, $|\mathbf{D}|$ is the total number of documents in the evaluation dataset, $|\mathbf{U}|$ is the total number of users who created gold-reference summaries that reflect their expected summaries (and thereby, their subjective preferences), and $|\mathbf{U}_{d_i}| \ (= |\mathbf{S}_{d_i}|)$ is the number of users who created gold-references for document $d_i$. $w$ is the divergence of the model-generated summary $s_{u_{ij}}$ (and the corresponding expected summary $u_{ij}$) from document $d_i$ itself in comparison to all the other versions. It helps to determine how much percentage (therefore, the softmax function) of the divergence (i.e., $\sigma(s_{u_{ij}}, s_{u_{ik}})$) should be considered for the calculation of DEGRESS. If $s_{u_{ij}}$ is farther than $s_{u_{ik}}$ w.r.t $d_i$ then DEGRESS($s_{u_{ij}}|(d_i, u_{ij})$) < DEGRESS($s_{u_{ik}}|(d_i, u_{ik})$), implying that $M_{\boldsymbol{\theta},u}$ is more responsive to the $k$-th reader. A lower value of DEGRESS($s_{u_{ij}}|(d_i, u_{ij})$) indicates that while reader-profiles are different, the generated summary $s_{u_{ij}}$ is very similar to other reader-specific summaries (or vice versa), and hence, is not responsive at the summary-level. The system-level DEGRESS and EGISES have been formulated as follows:

$$\text{DEGRESS}(M_{\boldsymbol{\theta},u}) = \frac{\sum\limits_{i=1}^{|\mathbf{D}|} \frac{\sum\limits_{j=1}^{|\mathbf{U}_{d_i}|} \text{DEGRESS}(s_{u_{ij}}|(d_i, u_{ij}))}{|\mathbf{U}_{d_i}|}}{|\mathbf{D}|} \quad (8)$$

### A.2   PERSEVAL: FORMULATION

As can be noted, the **DEGRESS formualtion does not enforce any penalty on accuracy drop**. To rectify this Dasgupta et al. (2024) proposed PerSEval. The design of PerSEval had two key goals: (i) to penalize models for poor accuracy, while simultaneously (ii) ensuring that the evaluation of responsiveness (i.e., DEGRESS) is not overshadowed by high accuracy. This penalty is referred to as the *Effective DEGRESS Penalty Factor* (EDP). If a model achieves 100% accuracy, no EDP will be applied, and the PerSEval score will equal the DEGRESS score. The following formulatiown of PerSEval guarantees these properties:

$$\text{PerSEval}(s_{u_{ij}}|(d_i, u_{ij})) = \text{DEGRESS}(s_{u_{ij}}|(d_i, u_{ij})) \times \text{EDP}(s_{u_{ij}}|(d_i, u_{ij}))$$

$$\text{where, EDP}(s_{u_{ij}}|(d_i, u_{ij})) = 1 - \frac{1}{1 + 10^{\alpha \geq 3} \cdot \exp\left(-(10^{\beta \geq 1} \cdot \text{DGP}(s_{u_{ij}}|(d_i, u_{ij})))\right)}, \quad (9)$$

$$\text{DGP}(s_{u_{ij}}|(d_i, u_{ij})) = \text{ADP}(s_{u_{i*}}|(d_i, u_{i*})) + \text{ACP}(s_{u_{ij}}|(d_i, u_{ij}))$$

Here, `ADP` is a document-level penalty due to a drop in accuracy for the best-performance of the model (i.e., the model-generated summary of document $d_i$ ($s_{u_{ij}}$) is closest to the corresponding reader's expected summary $u_{ij}$). `ADP` is formulated as follows:

$$\text{ADP}(s_{u_{i*}}|(d_i, u_{i*})) = \frac{1}{1 + 10^{\gamma \geq 4} \cdot \exp\left(-10 \cdot \frac{\sigma^*(s_{u_{i\bullet}}, u_{i\bullet})|d_i - \mathbf{0}}{(\mathbf{1} - \sigma^*(s_{u_{i\bullet}}, u_{i\bullet})|d_i) + \epsilon}\right)}$$

$$\text{where,} \; \sigma^*(s_{u_{i\bullet}}, u_{i\bullet})|d_i = \min_{j=1}^{|\mathbf{U}_{d_i}|} \sigma(s_{u_{ij}}, u_{ij})|d_i$$

$$\text{and } \{\epsilon : \text{An infinitesimally small number} \in (0, 1)\}$$

(10)

`ADP` ensures that even if the `DEGRESS` score is acceptable, a penalty due to accuracy drop can still be imposed as a part of `EDP`. `ADP`, however, fails to address the scenario where the best-case scenario is acceptable (i.e., accuracy is fairly high) but is rather an outlier case – i.e., for most of the other model-generated summary versions, there is a considerable accuracy drop. To address this issue, the second penalty component within `EDP` called *Accuracy-inconsistency Penalty* (`ACP`) was introduced which evaluates whether a model consistently performs w.r.t accuracy for a specific generated summary compared to its average performance. `ACP` is formulated as:

$$\text{ACP}(s_{u_{ij}}|(d_i, u_{ij})) = \frac{1}{1 + 10^{\gamma \geq 4} \cdot \exp\left(-10 \cdot \frac{\sigma(s_{u_{ij}}, u_{ij})|d_i - \sigma^*(s_{u_{i\bullet}}, u_{i\bullet})|d_i}{(\overline{\sigma}(s_{u_{i\bullet}}, u_{i\bullet})|d_i - \sigma^*(s_{u_{i\bullet}}, u_{i\bullet})|d_i) + \epsilon}\right)}$$

$$\text{where,} \; \overline{\sigma}(s_{u_{i\bullet}}, u_{i\bullet})|d_i = \frac{1}{|\mathbf{U}_{d_i}|} \sum_{j=1}^{|\mathbf{U}_{d_i}|} \sigma(s_{u_{ij}}, u_{ij})|d_i$$

(11)

The system-level `PerSEval` score is as follows:

$$\text{PerSEval}(M_{\boldsymbol{\theta}, u}) = \frac{\sum_{i=1}^{|\mathbf{D}|} \frac{\sum_{j=1}^{|\mathbf{U}_{d_i}|} \text{PerSEval}(s_{u_{ij}}|(d_i, u_{ij}))}{|\mathbf{U}_{d_i}|}}{|\mathbf{D}|}$$

(12)

The system-level `PerSEval` $\in [0, 1]$ and is bounded by the system-level `DEGRESS` score.

### A.2.1  PSE METRICS

**PerSEval-RG-SU4**  (or PSE-SU4) is the `PerSEval` variant that uses ROUGE-SU4 (Lin, 2004) as a distance metric (i.e., $\sigma$) in the `PerSEval` formula. PSE-SU4 has been reported to have high human-judgment correlation (Pearson's $r$: 0.6; Spearman's $\rho$: 0.6; Kendall's $\tau$: 0.51) (Dasgupta et al., 2024). The **ROUGE-SU4** score is based on *skip-bigrams*, which are pairs of words that appear in the same order within a sentence but can have up to four other words between them. The formula is as follows:

For a given generated summary $G$ and reference summary $R$, the ROUGE-SU4 score is calculated as:

**Skip-Bigram Recall ($R_{SU4}$):**

$$R_{\text{SU4}} = \frac{\text{Count of matching skip-bigrams between } G \text{ and } R}{\text{Total skip-bigrams in } R}$$

**Skip-Bigram Precision ($P_{SU4}$):**

$$P_{\text{SU4}} = \frac{\text{Count of matching skip-bigrams between } G \text{ and } R}{\text{Total skip-bigrams in } G}$$

**F1 Score ($F1_{SU4}$):** The F1 score is the harmonic mean of precision and recall:

$$F1_{\text{SU4}} = \frac{2 \times P_{\text{SU4}} \times R_{\text{SU4}}}{P_{\text{SU4}} + R_{\text{SU4}}}$$

Where:

- A **skip-bigram** consists of two words in the correct order but with zero to four words skipped in between.
- Matching skip-bigrams are counted between the generated summary and the reference summary.

The final **ROUGE-SU4** score is typically reported as the F1 measure, balancing precision and recall.

**PerSEval-JSD** (or PSE-JSD) is the `PerSEval` variant that uses the Jensen–Shannon Divergence (JSD) (Menéndez et al., 1997) as the distance metric $\sigma$ in the `PerSEval` formula. JSD is a smoothed and symmetric version of Kullback–Leibler divergence between the unigram (or n-gram) distributions of the generated summary $G$ and reference summary $R$. Its formulation is:

$$\mathrm{JSD}(P \,\|\, Q) \;=\; \tfrac{1}{2}\,\mathrm{KL}\!\left(P \,\middle\|\, M\right) \;+\; \tfrac{1}{2}\,\mathrm{KL}\!\left(Q \,\middle\|\, M\right) \quad \text{where} \quad M = \tfrac{1}{2}(P + Q) \tag{13}$$

here, $P$ and $Q$ are the normalized n-gram probability distributions of $G$ and $R$ respectively, and

$$\mathrm{KL}(P \| M) \;=\; \sum_x P(x) \, \log \frac{P(x)}{M(x)}.$$

We then define the divergence as: $\sigma_{\mathrm{JSD}}(G, R) = \mathrm{JSD}\!\left(P_G \| P_R\right)$ and plug $\sigma_{\mathrm{JSD}}$ into all occurrences of $\sigma$ in Equations equation 7–equation 12 to obtain PSE-JSD.

**PerSEval-Meteor** (or PSE-Meteor) uses the METEOR score (Banerjee & Lavie, 2005; Lavie & Agarwal, 2007) as the similarity metric; we convert it into a distance by $1 - \mathrm{METEOR}$. METEOR aligns unigrams (with synonymy, stem, and paraphrase matching) and combines precision, recall, and a fragmentation penalty. Its formulation is:

$$P = \frac{|\mathrm{matched\_unigrams}|}{|\mathrm{unigrams}(G)|}, \quad R = \frac{|\mathrm{matched\_unigrams}|}{|\mathrm{unigrams}(R)|}, \tag{14}$$

$$F_\alpha = \frac{P\,R}{\alpha\,P + (1 - \alpha)\,R}, \quad \alpha \in [0, 1], \tag{15}$$

$$\mathrm{Penalty} = \gamma \left( \frac{\#\mathrm{chunks}}{|\mathrm{matched\_unigrams}|} \right)^{\beta}, \quad \gamma, \beta > 0, \tag{16}$$

$$\mathrm{METEOR}(G, R) = (1 - \mathrm{Penalty}) \times F_\alpha. \tag{17}$$

We then set $\sigma_{\mathrm{Meteor}}(G, R) = 1 - \mathrm{METEOR}(G, R)$, and substitute $\sigma_{\mathrm{Meteor}}$ for $\sigma$ in Equations equation 7–equation 12 to yield PSE-Meteor.

### A.3  DETAILED BUILDUP OF PERDUCER ENCODER

**Initialization of u-Tier.** To enable Task 1 at the b-tier, we first initialise the user trajectory $\tau^{u_j}$ (u-tier). Each document ($d$) and summary ($s$) node receives an internal embedding from the SOTA KPE model *PromptRank* (220M param., 768-d) (Kong et al., 2023). Thus, for any behaviour triple $b_{u_j}^{(t_i)}$, the $hd$ and $tl$ nodes are seeded as $\mathbf{e}_{hd}^{(t_{i-1})}$ and $\mathbf{e}_{tl}^{(t_i)}$. KPE seeding aligns embeddings with central themes and outperforms SBERT baselines (Appendix E.2). The initial u-node embedding $\mathbf{e}_{u_j}^{(t_0)}$ is the title embedding of the first $d$-node, mitigating cold-start since no preference shift exists at $t_0$. Action-transition edges are seeded with a 4-d one-hot vector indicating *click*, *skip*, *genSumm*, or *summGen*.

#### A.3.1  B-TIER ENCODER

**The Base Model.** `PerDucer` has an RNN-styled recurrent base network of **b-cells** representing $\tau_b^{u_j}$, where each b-cell at time-step $t_i$ generates the b-node embedding $\mathbf{e}_{b_{u_j}}^{(t_i)}$. Each b-cell has three

sequential components– (i) the **head-cell**, (ii) the **action-cell**, and (iii) the **tail-cell**. The $t_i$-th head-cell fuses the incoming **behavior history** (i.e., the previous b-cell's **tail-cell content** $\mathbf{c}_{tl}^{(t_{i-1})}$) and the $hd$-node embedding $\mathbf{e}_{hd}^{(t_{i-1})}$ to generate the **head-cell content** $\mathbf{c}_{hd}^{(t_i)}$ as follows ($W_h, W_{hd}$ are learnable):

$$\mathbf{c}_{hd}^{(t_i)} = \tanh\left(W_h \cdot \mathbf{c}_{tl}^{(t_{i-1})} + \mathbf{b}_h\right) + \tanh\left(W_{hd} \cdot \mathbf{e}_{hd}^{(t_i)} + \mathbf{b}_{hd}\right) \tag{18}$$

The action-cell, representing one of the four possible transition actions, then fuses $\mathbf{c}_{hd}^{(t_i)}$ with the $a$-edge embedding $\mathbf{e}_a^{(t_i)}$ by projecting $\mathbf{c}_{hd}^{(t_i)}$ onto the action hyperplane[5] to generate **action-cell content** $\mathbf{c}_a^{(t_i)}$:

$$\mathbf{c}_a^{(t_i)} = \tanh\left(W_h \cdot \mathrm{proj}_{\mathbf{e}_a'^{(t_i)}} \mathbf{c}_{hd}^{(t_i)} + \mathbf{b}_{hd\perp a}\right) + \mathbf{e}_a'^{(t_i)}; \quad \text{where: } \mathbf{e}_a'^{(t_i)} = \tanh\left(W_a \cdot \mathbf{e}_a^{(t_i)} + \mathbf{b}_a\right) \tag{19}$$

Note that $W_a$ projects the 4-d 1-hot action-edge embedding onto a higher dimension equal to the head-cell content embedding. Finally, the tail-cell fuses $\mathbf{c}_a^{(t_i)}$ with the $tl$-node embedding $\mathbf{e}_{tl}^{(t_i)}$ by projecting back $\mathbf{c}_a^{(t_i)}$ onto the node-hyperplane to form the **tail-cell content** $\mathbf{c}_{tl}^{(t_i)}$:

$$\mathbf{c}_{tl}^{(t_i)} = \tanh\left(W_h \cdot \mathrm{proj}_{\mathbf{e}_{tl}'^{(t_i)}} \mathbf{c}_a^{(t_i)} + \mathbf{b}_{a\perp tl}\right) + \mathbf{e}_a'^{(t_i)}; \quad \text{where: } \mathbf{e}_{tl}'^{(t_i)} = \tanh\left(W_{tl} \cdot \mathbf{e}_{tl}^{(t_i)} + \mathbf{b}_{tl}\right) \tag{20}$$

The tail-cell content $\mathbf{c}_{tl}^{(t_i)}$ represents the content of the b-cell flowing onto the next b-cell. *The last b-cell content embedding represents* $\tau^{(u_j)}$. In the case of the first b-cell, the head-cell starts with the u-node embedding as input (see Section A.3; Figure 2). The $t_i$-th b-node embedding $\mathbf{e}_{b_{u_j}}^{(t_i)}$ is as follows:

$$\mathbf{e}_{b_{u_j}}^{(t_i)} = \tanh\left(W_b \cdot \mathbf{c}_{tl}^{(t_i)} + \mathbf{b}_b\right); \quad \text{where: } W_b \text{ is encoder header} \tag{21}$$

While $\mathbf{e}_{b_{u_j}}$ captures fine-grained behavior semantics at each step, it remains a *local representation* sensitive to the current behavior and near-past historical span. To model longer-term preference evolution and suppress spurious noise, we require a robust temporal aggregation mechanism. This motivates augmenting the b-tier architecture with a smooth yet adaptive history-aware encoding.

**History Aware Encoding via Decay-EMA.** Inspired by the damped-EMA module of the MEGA architecture proposed by Ma et al. (2023), we propose a **b-cell content-aware** *Decay-based Exponential Moving Average (D-EMA)* to capture the slow-drifting evolution of a user's interest over $\tau_b^{u_j}$. D-EMA recursively blends the current behavior representation with the smoothed history to form a *cumulative "snapshot"* representation $\mathbf{e}_{b_{u_j}^{\text{D-EMA}}}^{(t_{1:i})}$ as follows:

$$\mathbf{e}_{b_{u_j}^{\text{D-EMA}}}^{(t_{1:i})} = \alpha^{(t_i)} \odot \mathbf{e}_{b_{u_j}}^{(t_i)} + (1 - \alpha^{(t_i)} \odot \delta^{(t_i)}) \odot \mathbf{e}_{b_{u_j}^{\text{D-EMA}}}^{(t_{1:i-1})};$$

$$\text{where: } \alpha^{(t_i)} = \tanh\left(W_\alpha \cdot [\mathbf{e}_{b_{u_j}^{\text{D-EMA}}}^{(t_{i-1})}; \mathbf{e}_{b_{u_j}^{\text{D-EMA}}}^{(t_i)}] + \mathbf{b}_\alpha\right); \; \delta^{(t_i)} = \tanh\left(W_\delta \cdot [\mathbf{e}_{b_{u_j}^{\text{D-EMA}}}^{(t_{i-1})}; \mathbf{e}_{b_{u_j}^{\text{D-EMA}}}^{(t_i)}] + \mathbf{b}_\delta\right) \tag{22}$$

Here, $\alpha^{(t_i)}$ is a *learned decay coefficient* that is, unlike MEGA, *content-aware* since it modulates at every time-step based on the b-cell content inflow so far. In the same way, $\delta^{(t_i)}$ is a content-aware additional damping gate that modulates the degree of moving average, thereby making it possible for `PerDucer` encoder to give less weightage to near-past content on certain steps if required. This allows *adaptive control over how past behaviors influence the present* at $t_i$. However, the sequential blending inherently limits the ability to capture non-local dependencies - i.e., semantically similar behaviors that occur far apart in time but share conceptual themes or latent goals. In realistic user scenarios, preferences re-emerge or shift cyclically (e.g., returning to a topic after a gap), which D-EMA cannot model effectively.

**Contextualization of D-EMA via Self-Attention.** We augment D-EMA with *self-attention* mechanism to explicitly capture dependencies across all time steps, regardless of their temporal distance. This enables the model to contextualize the snapshot $\mathbf{e}_{b_{u_j}^{\text{D-EMA}}}^{(t_{1:i})}$ in terms of how each of the past cumulative behavior snapshots *independently* influences it. The updated contextualized embedding $\mathbf{e}_{b_{u_j}^{\text{c-EMA}}}^{(t_{1:i})}$

---

[5]The projection operation, inspired by TransH (Wang et al., 2014), distinguishes different cases of $(hd-tl)$-pair as determined by the type of transition-action, *particularly differentiating the click from the skip action.*

is generated using a single-head forward-masked Self Attention (**FM-Attn**) as[6]:

$$\mathbf{e}_{b_{u_j}^{\text{c-EMA}}}^{(t_{1:i})} = \phi_{\text{SiLU}} \left( W_{\text{c-EMA}} \cdot \left( \phi_{\text{SiLU}} \left( \mathbf{e'}_{b_{u_j}^{\text{D-EMA}}}^{(t_{1:i})} \right) + \mathbf{f}^{(t_i)} \odot \textbf{FM-Attn} \left( \mathbf{e}_{b_{u_j}^{\text{D-EMA}}}^{(t_{1:i})} \right) \right) + \mathbf{b}_{\text{c-EMA}} \right) \tag{23}$$

where: $W_{\text{c-EMA}}$ is learnable; and a forget gate at $t_i$: $\mathbf{f}^{(t_i)} = \phi_{\text{SiLU}} \left( W_f \cdot \mathbf{e'}_{b_{u_j}^{\text{D-EMA}}}^{(t_{1:i})} + \mathbf{b}_f \right)$

Although contextualized D-EMA provides a skip-gram modeling of discrete cumulative snapshots, the current time-step b-node information may get suppressed. We, therefore, add a calibrated residual (using the input gate $\mathbf{i}$) to generate the **content-aware MEGA (c-MEGA)** representation of $b_{u_j}^{t_i}$ as:

$$\mathbf{e}_{b_{u_j}^{\text{c-MEGA}}}^{(t_i)} = \mathbf{i}^{(t_i)} \odot \mathbf{e}_{b_{u_j}^{\text{c-EMA}}}^{(t_{1:i})} + (\mathbf{1} - \mathbf{i}) \odot \mathbf{e}_{b_{u_j}}^{(t_i)}; \quad \mathbf{i}^{(t_i)} = \sigma \left( W_i \cdot \mathbf{e'}_{b_{u_j}^{\text{D-EMA}}}^{(t_{1:i})} + \mathbf{b}_i \right) \tag{24}$$

**Predicting Next b-Node.** We apply a next node prediction header $W_{\text{pred}}$ on the last b-node embedding $\mathbf{e}_{b_{u_j}^{\text{c-MEGA}}}^{(t_l)}$ ($l$: length of the trajectory $\tau^{u_j}$) to predict the **query b-node embedding** at $t_{l+1}$ ($\mathbf{e}_{b_{u_j}^{\text{q}}}^{(t_{l+1})}$) as:

$$\mathbf{e}_{b_{u_j}^{\text{q}}}^{(t_{l+1})} = W_{\text{pred}} \cdot \mathbf{e}_{b_{u_j}^{\text{c-MEGA}}}^{(t_l)} + \mathbf{b}_{\text{pred}} \tag{25}$$

Note that the action-cell content of $\mathbf{e}_{b_{u_j}^{\text{c-MEGA}}}^{(t_l)}$ incorporates the embedding of the *genSumm* action on the query document $d_q^{(t_l)}$ which itself is incorporated within the tail-cell content (Figure 3).

# B DATASETS

## B.1 PENS DATASET

The PENS dataset (Ao et al., 2021) includes 113,762 news articles across 15 topics. Each article contains an ID, title (avg. 10.5 words), body (avg. 549 words), and category, with titles linked to WikiData entities. The dataset also includes user interaction data, such as impressions and click behaviors, combined with news bodies and headlines from the MIND dataset (Wu et al., 2020)

**PENS training set.** For training, 500k user-news impressions were sampled from June 13 to July 3, 2019. Each log records user interaction as [uID, tmp, clkNews, uclkNews, clkedHis], where 'clkNews' and 'uclkNews' represent clicked and unclicked news, and 'clkedHis' refers to the user's prior clicked articles, sorted by click time. The training data for `PerDucer`, as discussed in Section **??**, shows high preference shift. This inherently supports that personalizing UX is strongly dependent on the temporal dynamics of the user. The stats are in the table 9.

**PENS test set.** To create an offline testbed, 103 English-speaking students reviewed 1,000 headlines in stage-1, and then selected 50 articles, and created preferred headlines (i.e., expected gold-reference summaries) for 200 unseen articles in stage-2 (see Figure 4). Each article was reviewed by four participants. Editors checked for factual accuracy, discarding incorrect headlines. The high-quality remaining headlines serve as personalized gold-standard references in the PENS dataset.

## B.2 OPENAI (REDDIT) DATASET

The OpenAI (Reddit) dataset (Völske et al., 2017) comprises 123,169 Reddit posts collected from 29 distinct subreddits. This dataset provides both OpenAI-generated and human-written summaries and is organized into two splits: Comparisons, used for training and validation, and Axis, designated for validation and testing. A curated subset of 1,038 posts was processed by 13 different summarization policies, resulting in the generation of 7,713 summaries. These summaries underwent evaluation by 64 annotators who rated paired summaries based on selection preferences, confidence in their ratings, and dimensions such as accuracy, coherence, coverage, and overall quality. Notably, unlike datasets like PENS, these summaries are not linked to individual annotators or their reading histories, which means they lack elements of personalization and contextual user information.

---

[6] $\mathbf{e'}_{b_{u_j}^{\text{D-EMA}}}^{(t_{1:i})} = W_{\text{D-EMA}} \cdot \mathbf{e}_{b_{u_j}^{\text{D-EMA}}}^{(t_{1:i})} + \mathbf{b}_{\text{D-EMA}}$; $\mathbf{e'}_{b_{u_j}^{\text{D-EMA}}}^{(t_{1:i})}$ is the transformed residual; $W_{\text{D-EMA}}$ is learnable.

Table 6: Symbol Table

| Symbol | Meaning |
|---|---|
| UIG:$\langle N, E \rangle$ | User-Interaction Graph as a DAG with nodes $N$ and edges $E$ |
| $u_j^{(t_0)}$ | $j$-th user node (u-node) at initial time $t_0$ |
| $d^{(t_p)}$ | Document node (d-node) interacted at time-step $t_p$ |
| $s_j^{(t_q)}$ | User-specific expected summary node (s-node) at time-step $t_q$ |
| $a_d^{(t_p)}$ | Interaction edge on d-node at time-step $t_p$ (click/skip/genSumm) |
| $a_s^{(t_q)}$ | Follow-up edge from d-node to s-node at time-step $t_q$ (summGen) |
| $\tau^{u_j}$ | User trajectory (sequence of interactions) for user $u_j$ |
| $\mathcal{T}$ | Pool of user trajectories in the UIG |
| $\mathcal{T}_{\text{train}}, \mathcal{T}_{\text{test}}$ | Train and test splits of trajectory pool |
| $\mathcal{T}^{\text{PENS}}$ | Trajectory pool from PENS dataset (click/skip based) |
| $\mathcal{T}^{\text{PENS-D}}$ | Derived trajectory pool with test s-nodes incorporated |
| $\mathcal{T}^{\text{OAI}}$ | UIG-modeled trajectory pool from OpenAI-style dataset |
| $b_{u_j}^{(t_i)}$ | Behavior triple at time $t_i$: $\langle hd^{(t_{i-1})}, a^{(t_i)}, tl^{(t_i)} \rangle$ |
| $hd^{(t_{i-1})}$ | Head node of the behavior triple at time $t_{i-1}$ |
| $tl^{(t_i)}$ | Tail node of the behavior triple at time $t_i$ |
| $a^{(t_i)}$ | Action edge connecting head and tail at time $t_i$ |
| $\tau_b^{u_j}$ | b-tier trajectory for user $u_j$ (sequence of behavior triples) |
| $b_{(q,u_j)}$ | Query behavior triple to be predicted for user $u_j$ |
| $\mathbf{e}_{hd}^{(t_{i-1})}$ | Embedding of head node at time $t_{i-1}$ |
| $\mathbf{e}_{tl}^{(t_i)}$ | Embedding of tail node at time $t_i$ |
| $\mathbf{e}_{u_j}^{(t_0)}$ | Initial user embedding from first document title |
| $\mathbf{e}_a^{(t_i)}$ | Action edge embedding at time $t_i$ |
| $\mathbf{c}_{hd}^{(t_i)}$ | Head-cell content at time $t_i$ |
| $\mathbf{c}_a^{(t_i)}$ | Action-cell content at time $t_i$ |
| $\mathbf{c}_{tl}^{(t_i)}$ | Tail-cell content at time $t_i$ |
| $\text{proj}_{\mathbf{e}_a'^{(t_i)}}$ | Projection onto action hyperplane |
| $\mathbf{e}_a'^{(t_i)}$ | Projected action embedding |
| $W_h, W_{hd}, W_a$ | Learnable weight matrices for head/action embeddings |
| $\mathbf{b}_h, \mathbf{b}_{hd}, \mathbf{b}_a$ | Bias vectors for head and action projections |
| *clkNews*, *uclkNews* | Clicked and unclicked news entries in PENS |
| *genSumm*, *summGen* | Generation/follow-up edges for s-node interaction |

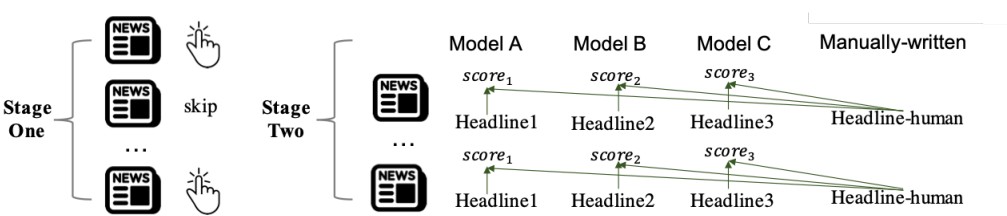

Figure 4: Stages of creation of testing dataset consisting of personalized headlines

## B.3 UIG Construction from Preference Datasets

In the parlance of UIG, preference datasets suitable for personalized summarization training and evaluation are of two categories– (i) those which can be directly modeled into a trajectory pool $\mathcal{T}$ (e.g., PENS dataset (Ao et al., 2021)) and (ii) those which lack user trajectories but contain discrete d-nodes, *model-generated* s-nodes (in contrast to user-generated s-nodes as per UIG definition), and *subjective* user feedback in the form of rating and the associated confidence score for that rating

Table 7: **MS/CAS PENS Dataset and Interaction Statistics**

| Characteristic | Dimension | Value |
|---|---|---|
| **Article Stats** | | |
| **General Stats** | # Topics | 15 |
| | # Articles | 113,762 |
| | Avg. Title Length | 10.5 words |
| | Avg. Body Length | 549 words |
| **Train Dataset Statistics** | | |
| **Interaction Data** | # User–News Impressions (anon.) | 500,000 |
| | # Users (anon.) | 445,000 |
| | Time Period | June 13–July 3, 2019 |
| | User Interaction Fields | [uID, tmp, clkNews, uclkNews, clkedHis] |
| **Test Dataset Statistics** | | |
| **Participant Stats** | # Participants | 103 |
| | Participant Category | English-speaking college students |
| | # Articles | 3,940 |
| | Browsed Headlines (Click + Skip) | 1,000 per participant |
| | Min. Interested (Click) Headlines | 50 per participant |
| **Gold Reference** **(Participant-written Headlines)** | Summarized Article Bodies | 200 per participant |
| | Avg. Summaries per Article | 4 |

Table 8: **OpenAI TL;DR (Reddit) Dataset Statistics**

| Characteristic | Dimension | Value |
|---|---|---|
| **Dataset Overview** | | |
| **General Stats** | # Reddit Posts | 123,169 |
| | # Subreddits (Domains) | 29 |
| | Policy-Generated Summaries | 115,579 |
| | Human-Written Summaries | Available |
| **Train + Validation Dataset Statistics** | | |
| **Article Stats** | # Reddit Posts | 21,111 |
| | # Policies | 81 |
| | # Generated Summaries | 107,866 |
| | # Annotators | 76 |
| | # Summary-Pairs Rated | 64,832 |
| **Validation Subset Statistics** | | |
| **Subset Details** | # Reddit Posts | 1,038 |
| | # Policies | 13 |
| | # Generated Summaries | 7,713 |
| | # Annotators | 32 |
| **Test Dataset (RLHF-Tuned Policies) Statistics** | | |
| **Evaluation Stats** | # Evaluated Policies | 4 |
| | # Evaluated Reddit Posts | 57 (out of 1,038) |
| | Evaluation Method | Indirect Benchmarking |
| **Annotation and Feedback** | | |
| **Feedback Collection** | Rating Scale | 1–7 |
| | Confidence Scale | 1–9 |
| | Avg. Ratings per Annotator | 1,176 |
| | Annotation Format | Summary-Pairs Selection |

(e.g. OpenAI-Reddit dataset (Völske et al., 2017)). We describe the UIG (i.e., the base u-tier) construction method for both types as follows:

**PENS-styled Datasets.** The construction of UIG is straightforward in the first case and is done in two steps. In the first step, *click* and *skip* interactions in the train dataset are mapped to document nodes (d-nodes) as incoming edges, forming the corresponding u-tier pool $\mathcal{T}$. As an example, for the PENS dataset, the *clkNews* interaction corresponds to a *click* edge and *uclkNews* to a *skip* edge, forming $\mathcal{T}^{\text{PENS}}$. However, PENS dataset lacks user-specific s-nodes (i.e., true interest evolution over

Table 9: **User-Interaction Graph Statistics** for our $\mathcal{T}_{\text{train}}^{\text{PENS-D}}$ and $\mathcal{T}_{\text{train}}^{\text{OAI}}$ only.

| Characteristic | $\mathcal{T}_{\text{train}}^{\text{PENS-D}}$ | $\mathcal{T}_{\text{train}}^{\text{OAI}}$ |
|---|---|---|
| # u-nodes (trajectories) | 150,000 | 45,000 |
| # d-nodes per trajectory | 123.7 | 36.92 |
| # s-nodes per trajectory | 15.10 | 11.44 |
| Average trajectory length | 129.8 | 48.37 |
| # Max. trajectory length | 200 | 50 |
| # Min. trajectory length | 5 | 25 |
| Rate of Topic Shift | 0.77 | 0.48 |

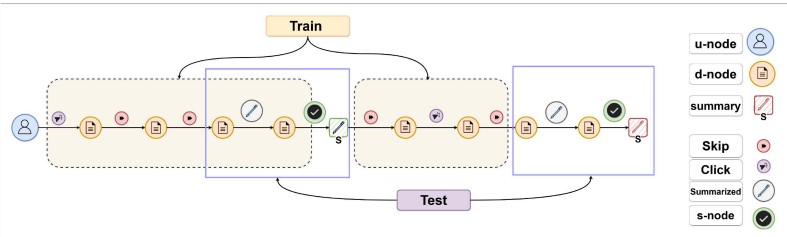

Figure 5: **UIG Construction**: Construction of User-Interaction Graph from preference datasets.

time), rendering $\mathcal{T}^{\text{PENS}}$ an *incomplete representation of the user dynamic preference*[7]. We address this issue in the second step, where we *incorporate* the s-nodes from the test dataset ($\mathcal{T}_{\text{test}}$) at their associated time-steps into $\mathcal{T}$ with the addition of *genSumm* and *summGen* edges, forming a derived (and more diverse) user-profile pool $\mathcal{T}^{\text{PENS-D}}$.

**OpenAI-styled Datasets.** For the second category of datasets, we first do a pre-construction classification of clicked and skipped d-nodes for every human rater $u_j$. This is done based on a simple heuristic of selecting those d-nodes as clicked which has at least one corresponding model-generated summary (note that there can be multiple models) that received a confidence score above a chosen threshold (in the case of OpenAI-Reddit we chose that to be 6 out of 9). We then select the best model-generated summary (i.e., one with the highest rating given by $u_j$) as the surrogate expected s-node for $u_j$. We then randomly sequence all such $(d - s)$-node pairs along with the skipped d-nodes to form $\tau^{u_j}$ (thereby $\mathcal{T}^{\text{OAI}}$). This method makes UIG-modeling *compatible with most summarization datasets that are not PENS-styled*.

## C  BASELINES

### C.1  BASELINE LLMS

**1. Zephyr 7B $\beta$.** Zephyr(Tunstall et al., 2023) is a 7B-parameter transformer model fine-tuned from Mistral-7B using Direct Preference Optimization (DPO) on publicly available and synthetic data. It removes some traditional alignment constraints to improve raw performance, achieving strong results on benchmarks like MT-Bench (7.34 vs. 6.86 for LLaMA2-70B-Chat). Zephyr is optimized for helpful dialogue and is openly available under an MIT license. Its design focuses on efficiency and high-quality responses without relying on reinforcement learning from human feedback.

**Mistral 7B.** Mistral-Instruct(Jiang et al., 2023) is a dense transformer model using grouped-query attention (GQA) and sliding window attention (SWA) to efficiently scale with long context inputs. Pretrained on around 2 trillion tokens, it delivers strong performance across NLP and coding benchmarks and surpasses larger models like LLaMA2-13B in many areas. It is fully open-source (Apache

---

[7]It is important to note that despite this, most recent frameworks train on $\mathcal{T}^{\text{PENS}}$ using history or document titles as "pseudo-targets" or via unsupervised learning (Ao et al., 2021; Song et al., 2023; Yang et al., 2023; Lian et al., 2025).

**Algorithm 1** UIG Construction

0: **function** CONSTRUCT_UIG(train_data, test_data, dataset_type)
0:     Initialize $\mathcal{T}_{\text{PENS}} \leftarrow \emptyset$, $\mathcal{T}_{\text{OAI}} \leftarrow \emptyset$
0:     **for** each user $u$ in train_data **do**
0:         Initialize $\tau_P^u \leftarrow \emptyset$, $\tau_{OAI}^u \leftarrow \emptyset$
0:         **for** each interaction in user $u$'s data **do**
0:             **if** dataset_type is PENS **then**
0:                 **if** interaction is *clkNews* **then**
0:                     Map to d-node with a *click* edge
0:                 **else if** interaction is *uclkNews* **then**
0:                     Map to d-node with a *skip* edge
0:                 **end if**
0:                 Append mapped d-node to $\tau_P^u$
0:             **else**
0:                 **if** model-generated summary rating $< 6$ **then**
0:                     Map to d-node with a *skip* edge
0:                 **else if** model-generated summary rating $> 6$ **then**
0:                     Map to d-node with a *click* edge
0:                 **end if**
0:                 **if** confidence for rating $=$ max **then**
0:                     Map to d-node with a *gensum* edge
0:                     Map to s-node with a *sumgen* edge
0:                 **end if**
0:                 Append mapped d-node to $\tau_{OAI}^u$
0:             **end if**
0:         **end for**
0:         **if** dataset_type is PENS **then**
0:             Add $\tau_P^u$ to $\mathcal{T}_{\text{PENS}}$
0:         **else**
0:             Add $\tau_{OAI}^u$ to $\mathcal{T}_{\text{OAI}}$
0:         **end if**
0:     **end for**
0:     **if** dataset_type is PENS **then**
0:         **for** each trajectory $\tau_P^u$ in $\mathcal{T}_{\text{PENS}}$ **do**
0:             Retrieve corresponding s-nodes from test_data at associated time-steps
0:             Insert s-nodes into $\tau_P^u$ using *genSumm* and *sumgen* edges
0:         **end for**
0:         $\mathcal{T}^{\text{PENS-D}} \leftarrow \mathcal{T}_{\text{PENS}}$
0:         **return** $\mathcal{T}^{\text{PENS-D}}$
0:     **else**
0:         **return** $\mathcal{T}_{\text{OAI}}$
0:     **end if**
0: **end function**=0

2.0) and includes an instruction-tuned variant, making it widely adopted for fine-tuning and deployment.

**LLaMA 2 13B.** LLaMA-2(Touvron et al., 2023) LLaMA 2 13B by Meta is a 13B-parameter autoregressive transformer trained on 2 trillion tokens of public data, with a context length of 4096. It supports chat via instruction tuning and RLHF. Though once state-of-the-art among open models, newer models like Mistral 7B now outperform it in many tasks. LLaMA 2 remains a strong, widely used foundation model with full documentation and open access under Meta's license.

**DeepSeek-R1 14B.** DeepSeek-R1(DeepSeek-AI et al., 2025) is a 14.8B-parameter model distilled from Qwen 2.5-14B, specifically optimized for math, code, and reasoning tasks. It was fine-tuned on 800K examples generated by a larger DeepSeek R1 model and is released under an MIT license.

Despite being smaller, it rivals much larger models on benchmarks like AIME and MATH, offering strong step-by-step reasoning while remaining efficient and open for further customization.

## C.2 BASELINE SLMs

**SmolLM2-1.7B.** SmolLM2 (Allal et al., 2025) is a lightweight language model with 1.7B parameters, designed for efficient performance on devices with limited resources. It offers fast inference and handles common NLP tasks well, making it a strong baseline for compact models. SmolLM2 was trained primarily on a mix of general-domain text tasks, including language modeling, next-word prediction, and basic text classification. The training involved supervised learning on curated datasets combined with unsupervised pretraining on large text corpora to build foundational language understanding while keeping the model compact.

**Qwen2.5-0.5B** Qwen2.5 (Qwen et al., 2025) is a smaller language model of 0.5B parameters, that balances scale and performance. It delivers better accuracy and versatility across NLP tasks, serving as a solid baseline for research and development without requiring massive computing power. Qwen2.5 was trained on a broader and more diverse set of tasks such as language modeling, question answering, summarization, and dialogue generation. It used a combination of large-scale unsupervised pretraining on extensive text data followed by supervised fine-tuning on specific downstream tasks to improve accuracy and contextual comprehension.

## C.3 BASELINE GENERIC SUMMARIZERS

**1. BigBirdPegasus.** BigbirdPegasus, proposed by (Zaheer et al., 2020) is an extension of Transformer based models designed specifically for processing longer sequences. It utilizes sparse attention, global attention, and random attention mechanisms to approximate full attention. This enables BigBird to handle longer contexts more efficiently and, therefore, can be suitable for summarization.

**2. SimCLS.** A Simple Framework for Contrastive Learning of Abstractive Summarization (Liu & Liu, 2021) uses a two-stage training procedure. In the first stage, a Seq2Seq model (Lewis et al., 2020) is trained to generate candidate summaries with MLE loss. Next, the evaluation model, initiated with RoBERTa is trained to rank the generated candidates with contrastive learning.

## C.4 BASELINE PERSONALIZED MODELS

**PENS-NRMS Injection-Type 1.** The PENS framework (Ao et al., 2021) generates personalized summaries by incorporating user embeddings along with the input news article. For this variant, user embeddings are derived using NRMS (Neural News Recommendation with Multi-Head Self-Attention) (Wu et al., 2019b), which includes a multi-head self-attention based news encoder to represent news titles, and a user encoder that captures browsing behavior through multi-head self-attention over clicked articles. Additive attention mechanisms are employed to highlight important words and articles. In Injection-Type 1, the NRMS user embedding is injected by initializing the decoder's hidden state, thereby directly influencing the summary generation process from the start.

**PENS-NRMS Injection-Type 2.** This variant also uses NRMS for user embedding, but personalization is introduced differently. Instead of initializing the decoder, the user embedding is injected into the attention mechanism of the PENS model. This modulates the attention weights over the news body, enabling the model to focus on content aligned with the user's preferences.

**PENS-NAML Injection-Type 1.** NAML (Neural News Recommendation with Attentive Multi-View Learning) (Wu et al., 2019a) generates news representations by attending over multiple views, including titles, bodies, and topic categories. The user encoder learns from interacted news and selects the most informative content for personalization. The resulting user embedding is integrated into the PENS decoder using Injection-Type 1, i.e., by initializing the decoder's hidden state.

**PENS-EBNR Injection-Type 1.** EBNR (Embedding-based News Recommendation) (Okura et al., 2017) models user preferences using an RNN over browsing histories to produce user em-

beddings. These embeddings are injected into the PENS model via Injection-Type 1 by initializing the decoder, thereby influencing the initial decoding steps with user-specific information.

**PENS-EBNR Injection-Type 2.** This configuration uses the same user encoder from EBNR but applies Injection-Type 2. Here, the user embedding is incorporated into the decoder's attention layers, allowing the model to personalize attention distributions over the news body during decoding.

**General Then Personal (GTP).** General Then Personal (GTP) (Song et al., 2023) is a two-stage framework for personalized headline generation. In stage-1, a Transformer-based encoder–decoder model is pre-trained on large-scale news article–headline pairs to learn robust, content-focused headline generation without personalization. In stage-2, a separate "headline customizer" refines the general headline by incorporating user-specific preferences, which are encoded as a control code by the user encoder TrRMIo. To bridge the gap between general generation and personalized refinement, GTP introduces two mechanisms: (i) **Information Self-Boosting (ISB)**, which reintroduces relevant content details from the article to prevent information loss during customization; and (ii) **Masked User Modeling (MUM)**, which randomly masks parts of the user embedding during training and reconstructs them, reducing the model's over-reliance on its general parameters.

**Signature Phrase.** Another line of personalization focuses on condensing a user's reading history into a collection of *signature phrases* (Cai et al., 2023). These phrases, derived through contrastive learning over news articles without annotated data, act as dynamic user profiles that adapt as interests evolve. Such phrases need not appear verbatim in the user's history but instead encode higher-level signals. Using these phrases, the model learns to generate personalized headlines that connect new articles with the user's inferred interests, yielding outputs that are engaging, relevant, and grounded in article content rather than drifting toward clickbait.

### C.5 Baseline Generic Summarizers

**BigBirdPegasus.** BigbirdPegasus, proposed by (Zaheer et al., 2020) is an extension of Transformer based models designed specifically for processing longer sequences. It utilizes sparse attention, global attention, and random attention mechanisms to approximate full attention. This enables BigBird to handle longer contexts more efficiently and, therefore, can be suitable for summarization.

**SimCLS.** A Simple Framework for Contrastive Learning of Abstractive Summarization (Liu & Liu, 2021) uses a two-stage training procedure. In the first stage, a Seq2Seq model (Lewis et al., 2020) is trained to generate candidate summaries with MLE loss. Next, the evaluation model, initiated with RoBERTa is trained to rank the generated candidates with contrastive learning.

## D Training Details

### D.1 Compute Resources

All preprocessing and embedding tasks were run on CPU-only machines, while model training utilized dedicated GPU servers. We utilized 16GB CPU cores for seeding embeddings with PromptRank on each node, for extracting keyphrase vocabulary with YAKE across all d-nodes, and for generating keyphrase ground-truth (distribution of keyphrases) for s-nodes using spaCy3.7. The training of each version of `PerDucer`, inferencing, and computing results were run with mixed-precision (FP16) training on NVIDIA L40 and L40S GPUs[8], alongside CPU-based preprocessing and data loading.

### D.2 Training

Model training comprised two sequential phases: first, `PerDucer` was trained end-to-end for 6 epochs, then the decoder was finetuned for 10 epochs. A batch size of 128 was used throughout,

---

[8]We gratefully acknowledge Lightning.ai for providing virtual compute resources using L40 and L40S GPUs.

**Algorithm 2** End-to-End Training Loop of `PerDucer`

```
0: function TRAIN_MODEL
0:    for each epoch do
0:        for each batch (B_hist, C_label) do
0:            L_ENC, L_KPE, L_total ← 0
0:            Initialize b_0^{c-MEGA} ← e_seed
0:            for t = 1 to n do
0:                b_t^{c-MEGA} ← ENCODE_BEHAVIOR(B_t)
0:                p̂_pos(t) ← SoftMax(W_pos b_t^{c-MEGA} + b_pos)
0:                L_pos ← − log p̂_pos(t)
0:                L_ENC ← L_ENC + L_pos
0:                b_next ← W_pred b_t^{c-MEGA} + b_pred
0:                p̂_kp ← W_mlp b_next + b_mlp
0:                L_KPE ← L_KPE − (1/k) Σ_{i=1}^{k} log p̂(kp_i)
0:            end for
0:            L_total ← α · L_ENC + (1 − α) · L_KPE
0:            optimizer.zero_grad()
0:            L_total.backward()
0:            optimizer.step()
0:        end for
0:    end for
0: end function=0
```

and optimization employed PyTorch's AdamW[9] with learning rate $1 \times 10^{-4}$ during encoder-only training and $1 \times 10^{-5}$ for joint fine-tuning, betas (0.9, 0.999), epsilon $1 \times 10^{-8}$, weight decay 0.01, a fixed learning rate policy, and dropout probability 0.1 on all self-attention and feed-forward layers. Total training steps were computed as $(N_\text{train}/128) \times 6$, where $N_\text{train}$ is the size of the training set. The vocabulary of keyphrases from training data is approximately 2252K, and the average number of keyphrases extracted from each s-node is 20. The total number of behaviors in the training data is 20700K. We used sampling softmax during training to speed up the training.

# E   DETAILED RESULTS

## E.1   PERSONALIZATION BOOSTING IN LLMS

We find that there is a consistent boost of personalization across all LLMs when `PerDucer`-guided keyphrases are supplied progressively with each build-up. The vanilla b-tier as Base Model shows effective boost of **25.3/18.1/22.5↑** wrt PSE-JSD/SU4/METEOR across all LLMs. $\mathcal{T}_\text{test}^\text{OAI}$ also shows boost of **13.14/9.34/15.25↑** when Base Model is used. D-EMA further boosts the results with best increase of **0.212/0.089/0.133↑** w.r.t. PSE-JSD/SU4/METEOR in PENS and **0.134/0.089/0.108↑** in OpenAI. FM-Attn on D-EMA results in slight boosting (sometimes drop) and further c-MEGA boosts the results in both PENS and OpenAI by approximately **0.105/0.154/0.174↑** and **0.143/0.151/0.157↑** in OpenAI. SBERT seeding boosts overall PSE in both datasets in terms of their with-history counterpart baselines, by an average of **0.0.34/0.0.36/0.42↑**. Detailed results are in Table 12.

## E.2   ABLATION STUDIES

**RQ-1 Ablation: Effect of the History Encoding Methods**   We find a steady boost over LLM baselines (2-shot user history) when the Base model is used to encode the user history $\tau_b^{u_j}$, with an average increase of **0.245/0.245/0.245↑** w.r.t. PSE-JSD/SU4/METEOR. Further, D-EMA on top of the Base model boosts the performance significantly, thereby indicating the importance of *historical snapshots* over purely *RNN-styled snowball* accumulation of histories. FM-Attn shows a slight boost, which might indicate that the long-term dependencies are captured. In fact, it is possible that long-term dependencies are already captured via D-EMA. Contextualization and residual connection

---

[9]AdamW implementation: `torch.optim.AdamW` (version 1.13.1)

Table 10: Learned Weights, Hyperparameters, and Dataset Statistics of `PerDucer`

| Parameter | Value / Shape |
|---|---|
| *Training Configuration* | |
| Batch size | 128 |
| Optimizer | AdamW (PyTorch-1.13.1) |
| Learning rate (encoder only) | $1 \times 10^{-4}$ |
| Learning rate (joint fine-tuning) | $1 \times 10^{-5}$ |
| Dropout | 0.1 |
| Epochs | 6 (1 encoder only + 5 joint) |
| Negative sampling | Enabled (10000 negs per pos) |
| Total training steps | $(N_{\text{train}}/128) \times 6$ |
| *Model Architecture* | |
| $d$ | 1560 |
| $\mathbf{E}_{\text{seed}}$ | 768 (220M params) |
| $\mathbf{a}$ | 4 |
| $W_{\text{pos}}$ | $20.7\text{M} \times 1560$ |
| $W_{\text{kp}}$ | $2.252\text{M} \times 1560$ |
| $W_{\text{pred}}$ | $1560 \times 1560$ |
| MLP before scoring | $1560 \rightarrow 512 \rightarrow 1560$ |
| *b-Tier Learned Weights* | |
| $W_h$ | $768 \times 768$ |
| $W_{hd}$ | $768 \times 768$ |
| $W_a$ | $4 \times 768$ |
| $W_{tl}$ | $768 \times 768$ |
| $W_b$ | $768 \times 768$ |
| *D-EMA and c-MEGA Gates* | |
| $W_{\alpha}$ | $1536 \times 768$ |
| $W_{\delta}$ | $1536 \times 768$ |
| $W_{\text{D-EMA}}$ | $768 \times 768$ |
| $W_f$ | $768 \times 768$ |
| $W_{\text{c-EMA}}$ | $768 \times 768$ |
| $W_i$ | $768 \times 768$ |
| *Data Preparation Statistics* | |
| $N_{\text{pos}}$ | 20.7M |
| $N_{\text{kp}}$ | 2.252M |
| Avg. keyphrases per node | 20 |

of FM-Attn and D-EMA lead to a significant boost again, indicating that both historical snapshots, as well as FM-Attn, are needed. SLMs reflect similar performance, and cross-domain experiments on OpenAI Reddit data further establish our point. Detailed results are discussed in Table 1.

**RQ-1 Ablation: Seed Embedding via SBERT.** We ablate on the quality of seed embedding using SBERT (Reimers & Gurevych (2019)) also to initialize the nodes. We find that there is an average drop of **11.21/7.32/10.86**↓ w.r.t. PSE-JSD/SU4/METEOR across all models. This supports out hypothesis that since the final downstream task of `PerDucer` is keyphrase extraction, PromptRank or similar type of model generates better quality of seed embeddings. Detailed results in Table 12.

**Ablation: Influence of LLM Temperature.** Varying temperature ($[0.2, 0.5, 0.8]$) shows that higher values degrade PSE (**0.13/0.16/0.2**↓, PSE-JSD/SU4/METEOR) as randomness increases, diluting key-phrase influence (Table 13).

**Human-Judgment Interpolation from OpenAI-Reddit dataset.** The interpolation of human judgment scores is performed by leveraging the OpenAI-Reddit dataset, which provides multiple human-

Table 11: **Accuracy Performance**: Comparison with Specialized and Vanilla Models.

| Category | Model | Rouge-SU4 | Rouge-L |
|---|---|---|---|
| Specialized (Personalized) | PENS-NAML-T1 | 13.12 | 21.62 |
| | PENS-EBNR-T1 | 12.16 | 20.73 |
| | PENS-EBNR-T2 | 12.41 | 20.82 |
| | PENS-NRMS-T1 | 13.15 | 20.75 |
| | PENS-NRMS-T2 | 13.64 | 21.03 |
| | GTP-TrRMIo | 21.91 | 28.31 |
| | SP-Individual | 19.54 | 25.18 |
| LLMs w/ 2-shot history) | LLaMA-13B | 18.31 | 29.54 |
| | Mistral-7B | 16.42 | 22.85 |
| | DeepSeek-14B | 19.57 | 29.72 |
| | Zephyr-7B | 18.45 | 26.45 |
| **PerDucer** | PerDucer+DeepSeek | **65.14** | **67.82** |
| | PerDucer+Mistral | **62.19** | **65.34** |
| | PerDucer+LLaMa | **63.55** | **67.16** |
| | PerDucer+Zephyr | **61.09** | **64.71** |

rated summaries for each article. For every article, the highest-rated human summaries which are 7 are designated as the *benchmark reference*. All candidate summaries, including the benchmark, are first embedded into a high-dimensional semantic space using a SentenceTransformer (Reimers & Gurevych, 2019) model. The semantic deviation between the benchmark embedding $V_b$ and any other summary embedding $V_o$ is quantified via the Root Mean Square Deviation (RMSD), which in this context is equivalent to the Euclidean distance:

$$\text{RMSD}(V_b, V_o) = \sqrt{\sum_{i=1}^{n}(b_i - o_i)^2} \, .$$

In practice, this computation is implemented efficiently using NumPy's linear algebra module, `np.linalg.norm`. The resulting RMSD values are then grouped according to the original human rating of each summary (e.g., $7/7$, $6/7$). By averaging the RMSD values within each rating group, we obtain a mapping between human-judged quality scores and embedding-space distances. Notably, the RMSD for summaries rated $7/7$ is not always zero, as there may exist multiple distinct summaries with a top score for the same article; while all such summaries are judged as equally high-quality by humans, their semantic embeddings can still differ due to variations in phrasing, emphasis, or lexical choices. These aggregated averages form the scoring thresholds used for interpolating human judgment in our evaluation framework.

## F  PROMPT TEMPLATE

As discussed in 4.3, we contrast our PerDucer-guided summarization with 0/2-shot user history and prompt-chaining w/user history-based summarization by LLMs. We provide a structured input by leveraging $\mathcal{T}^{\text{PENS-D}}$ as the user histories. On the other-hand, we just supply the main article along with the extracted keyphrases to the LLM to generate summaries. The detailed prompt structure is depicted in Figure 9.

## G  LICENSE AND USAGE STATEMENT

In this work, we utilize the following pre-trained large language models (PLMs) and small language models (SLMs):

- LLMs: DeepSeek-R1 14B (MIT License), Mistral-7B-Instruct (Apache 2.0), LLaMA2-13B (Llama 2 Community License), and Zephyr 7B ($\beta$) (MIT License).
- SLMs: SmolLM2 1.7B (Apache 2.0) and Qwen2.5 0.5B (Apache 2.0).

Table 12: **Performance of LLMs when prompted with 2-shot user history, vs. when guided with different versions of `PerDucer` encoder. Observation-1**: *All* `PerDucer` *versions beat the baseline 2-shot prompting across all LLMs*; **Observation-2**: *c-MEGA outperforms all other versions* of `PerDucer`, *thereby indicating the need of Residual Fusion of D-EMA with D-EMA+FM-Attn*; **Observation-3**: *Although DeepSeek outperforms all other models, the significant performance of smaller models at par with LLMs indicate that even SLMs can perform equivalent to LLMs when the task is narrowed down*; **Observation-4**: *Seed embeddings with SBERT results in performance drop across all the models, thereby establishing the fact that PromptRank seeding is a superior seeding since the final task is keyphrase extraction.* [*SLMs are not benchmarked with user history due to lower context size.]

| Context-Source | LLM/SLM | MS/CAS PENS Test | | | OpenAI Reddit Test | | |
|---|---|---|---|---|---|---|---|
| | | PSE-JSD | PSE-SU4 | PSE-METEOR | PSE-JSD | PSE-SU4 | PSE-METEOR |
| 2-shot History | Mistral-7B | 0.235 | 0.087 | 0.084 | 0.226 | 0.088 | 0.103 |
| | DeepSeek-R1 | 0.248 | 0.094 | 0.097 | 0.243 | 0.095 | 0.109 |
| | Zephyr-7B-$\beta$ | 0.231 | 0.085 | 0.086 | 0.214 | 0.087 | 0.104 |
| | LLaMA-13B | 0.227 | 0.078 | 0.081 | 0.232 | 0.093 | 0.107 |
| | Qwen2.5-0.5B | NA* | NA* | NA* | NA* | NA* | NA* |
| | smolLM2-1.5B | NA* | NA* | NA* | NA* | NA* | NA* |
| B-tier Vanilla | Mistral-7B | 0.484 | 0.275 | 0.319 | 0.343 | 0.177 | 0.258 |
| | DeepSeek-R1 | **0.513** | **0.292** | **0.322** | 0.377 | **0.202** | 0.244 |
| | Zephyr-7B-$\beta$ | 0.505 | 0.281 | **0.322** | 0.341 | 0.171 | 0.153 |
| | LLaMA-13B | 0.435 | 0.269 | 0.303 | 0.356 | 0.187 | **0.266** |
| | Qwen2.5-0.5B | 0.347 | 0.238 | 0.264 | 0.282 | 0.137 | 0.154 |
| | smolLM2-1.5B | 0.431 | 0.284 | 0.338 | **0.362** | 0.200 | 0.231 |
| D-EMA | Mistral-7B | 0.597 | 0.359 | 0.425 | 0.437 | 0.285 | 0.338 |
| | DeepSeek-R1 | **0.602** | **0.362** | **0.429** | **0.453** | 0.246 | 0.276 |
| | Zephyr-7B-$\beta$ | 0.566 | 0.352 | 0.401 | 0.422 | 0.244 | **0.518** |
| | LLaMA-13B | 0.482 | 0.361 | 0.417 | 0.445 | **0.294** | 0.366 |
| | Qwen2.5-0.5B | 0.559 | 0.327 | 0.397 | 0.416 | 0.221 | 0.276 |
| | smolLM2-1.5B | 0.599 | 0.360 | 0.427 | 0.446 | 0.240 | 0.288 |
| D-EMA + FM-Attn | Mistral-7B | 0.572 | 0.382 | 0.445 | 0.473 | 0.314 | 0.386 |
| | DeepSeek-R1 | **0.583** | **0.390** | **0.453** | **0.501** | 0.326 | 0.284 |
| | Zephyr-7B-$\beta$ | 0.591 | 0.364 | 0.433 | 0.467 | 0.293 | 0.346 |
| | LLaMA-13B | 0.509 | 0.379 | 0.439 | 0.482 | **0.338** | **0.385** |
| | Qwen2.5-0.5B | 0.522 | 0.333 | 0.384 | 0.448 | 0.273 | 0.302 |
| | smolLM2-1.5B | 0.544 | 0.383 | 0.443 | 0.502 | 0.329 | 0.348 |
| C-MEGA | Mistral-7B | 0.676 | 0.524 | 0.604 | 0.612 | 0.452 | 0.503 |
| | DeepSeek-R1 | **0.710** | **0.543** | **0.627** | **0.632** | **0.473** | **0.524** |
| | Zephyr-7B-$\beta$ | 0.695 | 0.530 | 0.607 | 0.624 | 0.471 | 0.518 |
| | LLaMA-13B | 0.685 | 0.533 | 0.614 | 0.627 | 0.473 | 0.521 |
| | Qwen2.5-0.5B | 0.652 | 0.467 | 0.585 | 0.584 | 0.434 | 0.458 |
| | smolLM2-1.5B | 0.700 | 0.536 | 0.615 | 0.628 | 0.470 | 0.521 |
| C-MEGA (SBert Seed) | Mistral-7B | 0.622 | 0.472 | 0.497 | 0.521 | 0.357 | 0.435 |
| | DeepSeek-R1 | **0.642** | **0.487** | **0.538** | **0.545** | **0.392** | **0.446** |
| | Zephyr-7B-$\beta$ | 0.579 | 0.453 | 0.523 | 0.503 | 0.322 | 0.374 |
| | LLaMA-13B | 0.533 | 0.431 | 0.482 | 0.533 | 0.356 | 0.421 |
| | Qwen2.5-0.5B | 0.513 | 0.402 | 0.463 | 0.474 | 0.316 | 0.354 |
| | smolLM2-1.5B | 0.576 | 0.441 | 0.482 | 0.495 | 0.375 | 0.363 |

All models are used according to their respective licenses and terms provided by their original creators. Proper attribution is given to each model's developers as cited in our references.

We also use the following datasets:

- **MS/CAS PENS dataset:** We comply with the dataset's terms of use, which is derived from the Microsoft Research License (`https://github.com/msnews/MIND/blob/master/MSR%20License_Data.pdf`).
- **OpenAI Reddit dataset:** We comply with the MIT License specifications as set by OpenAI (`https://github.com/openai/summarize-from-feedback/blob/master/LICENSE`)

We have ensured that all datasets and models are used responsibly, respecting privacy, consent, and ethical guidelines. When applicable, data is anonymized and handled according to the ethical standards set forth by NeurIPS.

Table 13: **Ablation with temperature values across different LLMs**.

| Temperature | LLMs | PSE-Scores | | |
|---|---|---|---|---|
| | | **PSE-JSD** | **PSE-SU4** | **PSE-METEOR** |
| 0.2 | Mistral-7B | 0.676 | 0.524 | 0.604 |
| | DeepSeek-R1 | **0.710** | **0.543** | **0.627** |
| | Zephyr-7B-$\beta$ | 0.694 | 0.53 | 0.607 |
| | LLaMA-13B | 0.685 | 0.533 | 0.614 |
| 0.5 | Mistral-7B | 0.581 | 0.415 | 0.463 |
| | DeepSeek-R1 | 0.651 | 0.476 | 0.529 |
| | Zephyr-7B-$\beta$ | 0.608 | 0.384 | 0.431 |
| | LLaMA-13B | 0.593 | 0.489 | 0.496 |
| 0.8 | Mistral-7B | 0.502 | 0.314 | 0.325 |
| | DeepSeek-R1 | 0.516 | 0.322 | 0.365 |
| | Zephyr-7B-$\beta$ | 0.472 | 0.304 | 0.353 |
| | LLaMA-13B | 0.497 | 0.319 | 0.368 |

Table 14: **RMSD w.r.t. gold reference summaries and approximated HJ Rating from annotated OpenAI-Reddit dataset for Different Models**

| Model | RMSD | HJ Rating |
|---|---|---|
| EBNR-1 | 0.9319 | 2 |
| EBNR-2 | 0.9378 | 2 |
| NAML-1 | 0.9260 | 2 |
| NRMS-1 | 0.9108 | 2 |
| NRMS-2 | 0.9187 | 2 |
| GTP | 0.9382 | 2 |
| SP | 0.8814 | 3 |
| Mistral (2-shot) | 0.7913 | 5 |
| DeepSeek (2-shot) | 0.7786 | 5 |
| PerDucer + DeepSeek | 0.3361 | 7 |
| PerDucer + Mistral | 0.3418 | 7 |

**0-shot w/ history**

**User History**

List of Articles clicked/Skipped/Summarized by user:
<Doc1: click>, <Doc2: click>, <Doc3: skip>,
<Doc4: Summarized as Sum1>..........

**Task**

Given Query Doc <doc_content>

Generate a Headline by considering the user's history as the indicator to their interests, where click denotes positive interest, skip denotes negative interest and summarized indicates focus on that topic. Return the headline in this format: Headline: {output}

Figure 6: **0-shot prompting** Patel et al. (2024))

**2-shot w/ history**

**User History**

List of Articles clicked/Skipped/Summarized by user:
<Doc1: click>, <Doc2: click>, <Doc3: skip>, <Doc4: Summarized as Sum1>...........

**2 shot examples**

[doc_content]
[Personalized Headline]: Rewritten_Titles by User
-----------------------------------------
[Doc Content]
[Personalized Headline]: Rewritten_Titles by User

**Task**

Given Query Doc <doc_content>

Generate a Headline by considering the user's history as the indicator to their interests, where click denotes positive interest, skip denotes negative interest and summarized indicates focus on that topic. Return the headline in this format: Headline: {output}

Figure 7: **0-shot prompting** Patel et al. (2024))

**Prompt-Chaining w/ history**

**User History**

List of Articles clicked/Skipped/Summarized by user one by one:
<Doc1: click>

**Task**

Given doc and action performed <doc1_content, click>

Generate a list of interested keyphrases, topics, and preferences for the user.

**Output**

Interest: <topic1, topic2, topic3>
Keyphrases: <phrase1, phrase2, phrase3>

**User History**

List of Articles clicked/Skipped/Summarized by user one by one:
<Doc2: skip>

**Task**

Given doc and action performed <doc1_content, click>, and the user preference output

Update a list of interested keyphrases, topics, and preferences for the user.

Figure 8: **chain-based prompting**

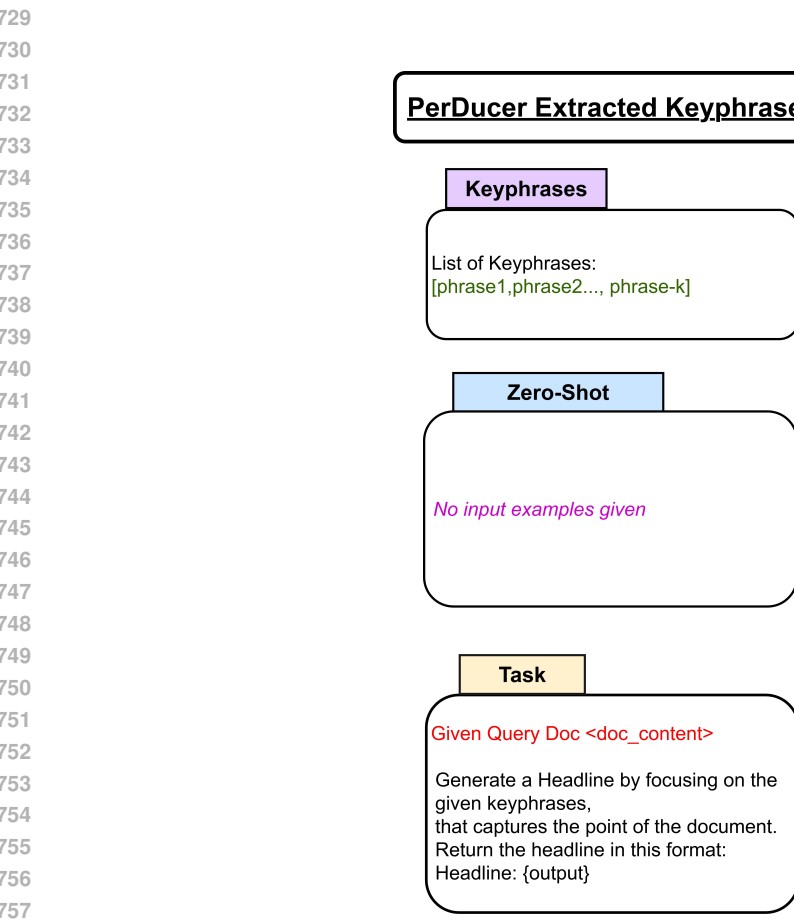

Figure 9: **PerDucer Top-$k$ Key-Phrase Guidance**: Cue injection in LLM/SLM

Table 15: **Sequential Recommendation Task on Perducer+Language Model on Amazon Beauty**. We utilize Amazon Beauty dataset (public: `https://cseweb.ucsd.edu/~jmcauley/datasets/amazon/links.html`) for sequential recommendation task. We find as a set of preliminary results that `PerDucer` can boost the sequential recommendation task too, performing at par with SOTA sequential recommenders, with the best performing DeepSeek (and SmolLM2) with just 0.03 and 0.01 behind w.r.t nDCG@20. This shows that `PerDucer` framework can boost any kind of personalization when fitted/adapted with LLMs (or similar models).

| Model | nDCG@20 |
|---|---|
| BSARec Shin et al. (2024) | **0.07** |
| TiM4Rec Fan et al. (2024) | 0.05 |
| Perducer+DeepSeek | **0.04** |
| Perducer+Mistral | 0.03 |
| Perducer+SmolLM2 | **0.04** |

