# OpenReview forum: "PerDucer: Personalization Inducer for Text Summarizers via User Preference Prediction"
_ICLR.cc/2026/Conference — ICLR 2026 Conference Withdrawn Submission_

### Official Review · Reviewer_Ww6y · 2025-10-18

**Soundness:** 3
**Presentation:** 3
**Contribution:** 3
**Rating:** 4
**Confidence:** 4

**Summary:**

PerDucer reformulates personalized summarization as keyphrase-guided generation. It predicts a user's next interaction from dynamic preference histories (modeled as a temporal User-Interaction Graph) and extracts ranked key-phrases as cues to boost frozen summarizers, addressing LLMs' limitations with long contexts.

Contributions

- Model-Agnostic Boosting: Improves SOTA LLMs (e.g., avg. 0.47↑ on PerSEval metrics), elevates SLMs to near-LLM performance (SmolLM2 at 98.6% of DeepSeek-14B), and pushes vanilla models (e.g., BigBird-Pegasus) beyond specialized baselines like GTP (up to 0.20↑).

- Empirical Validation: Evaluated on PENS (real histories) and synthetic OpenAI-Reddit datasets using PerSEval variants, confirming effectiveness of key-phrase injection for evolving preferences.

**Strengths:**

Originality

The paper introduces a novel reformulation of personalized summarization as keyphrase-guided generation, leveraging a temporal User-Interaction Graph (UIG) to model dynamic user preferences—distinguishing it from prior static persona-based approaches. By predicting user interactions and extracting ranked keyphrases as cues for frozen summarizers, PerDucer creatively combines sequential recommendation techniques (e.g., Decay-EMA and forward-masked attention) with keyphrase extraction, addressing LLM limitations in handling long histories without fine-tuning. This model-agnostic booster extends ideas from works like GTP and Signature-Phrase but innovates by incorporating action-specific edges (click/skip/summarize) and bi-level UIG abstraction, enabling SLMs to rival LLMs.

Quality

PerDucer's technical design is robust, with a clear encoder-decoder pipeline validated through rigorous experiments on PENS (real-user histories) and OpenAI-Reddit (synthetic). It demonstrates consistent gains (avg. 0.47↑ on PerSEval metrics) across four LLMs and two SLMs, outperforming history-prompt baselines and specialized models like GTP by up to 0.20↑. Ablations on architecture (c-MEGA vs. D-EMA), seeding (PromptRank > SBERT), and temperature affirm the framework's soundness.

Clarity

The manuscript is well-structured, with precise definitions (e.g., UIG formulation in Sec. 3.1), illustrative figures (e.g., PerDucer pipeline), and mathematical derivations (e.g., c-MEGA equations). Explanations of challenges like the ICOPERNICUS Paradox are accessible, and appendices provide detailed notations, algorithms, and prompts. While dense in technical terms, the progression from motivation to evaluation ensures logical flow, making it readable for NLP researchers.

Significance

By enabling efficient personalization without retraining, PerDucer has broad implications for resource-constrained applications, elevating SLMs to near-LLM performance (e.g., SmolLM2 at 98.6% of DeepSeek-14B) and extending to recommendations (preliminary Amazon Beauty results). It advances handling of evolving preferences in multi-aspect content, potentially reducing bias in AI systems and improving user engagement in news, search, or e-commerce.

**Weaknesses:**

Methodology and Novelty

While PerDucer's reformulation of personalization as keyphrase-guided summarization via a User-Interaction Graph (UIG) is innovative in incorporating action-specific edges (e.g., click, skip, summarize) and bi-level abstractions, it overlaps significantly with prior graph-based personalization frameworks, potentially limiting its claimed originality. For instance, the UIG's temporal modeling of user trajectories echoes the graph-based user modeling in GTP (Song et al., 2023) , which also derives latent controls from user histories for personalized headline generation, including decoupling general and personal components. Although PerDucer differentiates by emphasizing short- vs. long-term dependencies via Decay-EMA and self-attention, it does not sufficiently benchmark against GTP's editing controls or demonstrate why UIG's action distinctions yield superior handling of dynamic preferences—e.g., through ablation on action types vs. GTP's latent edits. To strengthen novelty, the authors could extend comparisons to include recent RL-based approaches like PLUS (Ao et al., 2025) , which learns personalized summaries from user information via reinforcement learning, by integrating RL for keyphrase ranking or showing UIG's advantages in preference drift scenarios.
The encoder's complexity, with stacked b-cells, content-aware Decay-EMA (D-EMA), and forward-masked self-attention (FM-Attn), risks over-engineering for the task, as simpler trajectory encoders (e.g., TrRMIo in GTP) achieve comparable personalization without such layers. Empirical evidence from sequential recommendation literature (e.g., Xia et al., 2022, cited in the paper) supports hierarchical abstractions, but the paper's ablations (e.g., c-MEGA vs. D-EMA) are limited to internal components; adding baselines like graph neural networks (GNNs) for UIG encoding, as in personalized graph summarization (Koutra et al., 2022) , would clarify if the RNN-style stack is necessary or if a GNN could simplify while maintaining performance.

Experimental Evaluation

The reliance on PENS (Ao et al., 2021) as the primary real-world dataset is a key limitation, given its small scale (averaging 13.6 topics per user and ~15K test rows) and lack of diverse user signals like reading time or content engagement, which restricts generalizability to real evolving preferences. As noted in recent critiques (e.g., Vansh et al., 2024) , PENS's annotations are biased toward English-speaking users and overlook fine-grained interactions, potentially inflating PerSEval scores on synthetic dynamics. The synthetic OpenAI-Reddit derivation exacerbates this, as it imposes artificial temporal orders on non-sequential data (Völske et al., 2017), failing to capture authentic preference shifts—e.g., cyclical themes as in the Alice example. To address this, incorporate larger, more diverse datasets like PersonalSum (Zhang et al., 2024, cited) or the recent User-Subjective Guided dataset (Vansh et al., 2024) , which includes user-subjective signals (e.g., plot/structure preferences), and report cross-dataset transfer learning results to validate robustness.
Human evaluation is interpolated but undetailed in the main text (likely in appendices), relying on alignment claims with PerSEval without raw inter-annotator agreement or preference rankings from diverse users. This is insufficient for subjective tasks, as PerSEval variants (JSD/SU4/METEOR) penalize accuracy drops but may not capture nuance in multi-aspect documents (Dasgupta et al., 2023) . Conduct targeted human studies (e.g., A/B tests on 100+ users) comparing PerDucer-boosted summaries to baselines on metrics like perceived personalization and uniqueness, as in PLUS evaluations .
Scalability testing is absent for long histories (e.g., >200 steps in PENS train), where RNN-style encoders could suffer quadratic attention costs despite FM-Attn masking. Benchmark on extended synthetic trajectories (e.g., 500+ steps) and report inference times, drawing from efficiency critiques in long-context LLMs (Liu et al., 2024, cited).

Significance and Broader Impact

While PerDucer's model-agnostic boosting aligns with sustainable AI by avoiding fine-tuning, its focus on news-like domains (PENS/OpenAI-Reddit) limits applicability to diverse areas like e-commerce reviews or scientific texts, where graph-based personalization has shown promise (e.g., personalized review summarization, Li et al., 2024) . Preliminary Amazon Beauty results (mentioned) are promising but undetailed; expand to full experiments on a review dataset to demonstrate cross-domain transfer.

**Questions:**

Detailed Comparison with GTP and Action Distinctions: The paper positions PerDucer as advancing beyond GTP (Song et al., 2023) by modeling temporal dependencies and distinguishing user actions (e.g., click, skip, summarize) in the UIG. However, it's unclear how UIG's action-specific edges empirically outperform GTP's TrRMIo encoder, which also handles trajectories but without explicit action types. Could you provide an ablation study comparing UIG with a GTP-style latent edit encoder on the same metrics? This could clarify the novelty and potentially strengthen my view on PerDucer's superiority in capturing dynamic preferences.

Integration or Comparison with Reinforcement Learning Approaches: Recent works like those in personalized text generation (e.g., as referenced in "Comparative Personalization for Multi-document Summarization" arXiv:2509.21562v1, 2025) explore RL for aligning summaries with user feedback. Given PerDucer's focus on predicting next behaviors, have you considered incorporating RL to refine keyphrase ranking based on simulated user rewards?

Details on Human Evaluation and Inter-Annotator Agreement: The paper claims PerSEval aligns with human judgment but provides no specifics on human studies in the visible sections. If such evaluations exist (e.g., in appendices), could you share details like the number of annotators, inter-annotator agreement (e.g., Kappa scores), and A/B preference tests on ~100 users comparing PerDucer-boosted summaries to baselines? This would clarify the subjective validity of gains (e.g., 0.47↑ on PerSEval) and could resolve my concern that automatic metrics alone are insufficient for personalization tasks.

Scalability Testing for Long Histories: The encoder's RNN-style stack with self-attention may incur high costs for very long trajectories (>200 steps in training), but no scalability results are reported. Could you provide benchmarks on extended synthetic trajectories (e.g., 500+ steps) including inference times and memory usage?

Expansion of Preliminary Cross-Domain Results: The paper hints at preliminary Amazon Beauty results for recommendations but provides no details. Could you share full experimental outcomes on a review dataset (e.g., gains over baselines) to demonstrate transfer beyond news domains? This would broaden the significance, especially for e-commerce, and could mitigate my view of the work as domain-limited.

---

### Official Review · Reviewer_Zxxh · 2025-10-20

**Soundness:** 2
**Presentation:** 3
**Contribution:** 2
**Rating:** 4
**Confidence:** 4

**Summary:**

The authors propose PerDucer, a model-agnostic booster that reframes personalization as keyphrase-guided summarization, avoiding costly fine-tuning. PerDucer first models user interaction histories as a temporal User-Interaction Graph, leveraging a novel c-MEGA encoder which integrates damped exponential moving average and self-attention to capture preference evolution and predict next-step behavior embeddings. Experiments on datasets like PENS (personalized news) and OpenAI-Reddit demonstrate that PerDucer consistently boosts performance.

**Strengths:**

- The framework acts as a ​​booster​​ for frozen summarizers (both LLMs and SLMs), avoiding resource-intensive fine-tuning. This flexibility allows integration with diverse models, from BigBird-Pegasus to Mistral-7B, without architectural changes.
- PerDucer trained on OpenAI-Reddit (non-news) transfers well to PENS (news), demonstrating adaptability to diverse domains without retraining.

**Weaknesses:**

- For users with short or sparse interaction trajectories (e.g., cold-start scenarios), PerDucer’s UIG may lack sufficient signals for reliable preference prediction.
- This paper lacks comarison with other personalization modeling methods in experiments. For example, those agentic memory approaches [1],
- Note that the DeepSeek-R1 in Table 1 refers to DeepSeek-R1-Distill-Qwen-14B, but the authors did not explain this abbreviation in caption, making the results misleading. Readers might think you use the 671B DeepSeek-R1.
- The tables in this paper are very small. I feel hard to read those small fonts.


[1] Zhang, Weizhi, et al. "Personaagent: When large language model agents meet personalization at test time." arXiv preprint arXiv:2506.06254 (2025).

**Questions:**

- How does the c-MEGA encoder in PerDucer overcome the limitations of standard RNN-based approaches in modeling long-term user preference evolution?​
- Why not evaluate your methods on the real SOTA open-source LLMs like DeepSeek-V3.1 [1] and GLM-4.5 [2]?



[1] Liu, Aixin, et al. "Deepseek-v3 technical report." arXiv preprint arXiv:2412.19437 (2024).

[2] Zeng, Aohan, et al. "Glm-4.5: Agentic, reasoning, and coding (arc) foundation models." arXiv preprint arXiv:2508.06471 (2025).

---

### Official Review · Reviewer_PMq9 · 2025-10-29

**Soundness:** 1
**Presentation:** 1
**Contribution:** 1
**Rating:** 2
**Confidence:** 4

**Summary:**

This paper focuses on modeling the evolution of user preferences for abstractive summaries in the context of personalized summarization. To this end, the authors propose a novel graph structure called the User-Interaction Graph, in which nodes represent users, documents, and summaries, while edges encode user interactions. Based on this graph, they design an RNN-style graph network to encode user trajectories and decode them into key phrases that capture user preferences. These key phrases are then incorporated into the prompt of a large language model (LLM) to improve generation quality. Experiments were carried out on PENS and OpenAI-Reddit dataset. Results with n-gram based metrics suggest that their approach improves the performance of SOTA LLMs, raises the performance of SLMs, and surpassing the performance of specialized systems with vanilla summarizers.

**Strengths:**

The paper focuses on the well-motivated and useful task of personalized summarization. It specifically focuses on modeling fine-grained, dynamically evolving user preferences at the sub-topic level through user interactions.

**Weaknesses:**

* The manuscript requires thorough proofreading. Numerous grammatical errors and incomplete sentences make it difficult to follow.
* The selected personalized summarization models, which range from 2017 to 2023, lack representativeness in terms of recency. This weakens the persuasiveness of the experimental results.
* The evaluation highly relied on n-gram based matching metrics, without considering more advanced semantic-based metrics.

**Questions:**

*  The manuscript requires language polishing and reorganization to enhance its clarity. Additionally, the inclusion of more illustrative examples and case studies would be helpful.
* It would be valuable to include comparisons with recent LLM Agent architectures, especially those featuring advanced memorization and reflection mechanisms [1].
* Expanding the evaluation framework to include semantic-based metrics like G-Eval [2] is encouraged.

[1] Wu, Yaxiong, Sheng Liang, Chen Zhang, Yichao Wang, Yongyue Zhang, Huifeng Guo, Ruiming Tang, and Yong Liu. "From human memory to ai memory: A survey on memory mechanisms in the era of llms." arXiv preprint arXiv:2504.15965 (2025).
[2] Liu, Yang, Dan Iter, Yichong Xu, Shuohang Wang, Ruochen Xu, and Chenguang Zhu. "G-Eval: NLG Evaluation using Gpt-4 with Better Human Alignment." In Proceedings of the 2023 Conference on Empirical Methods in Natural Language Processing, pp. 2511-2522. 2023.

---

### Official Review · Reviewer_8Fz9 · 2025-11-02

**Soundness:** 1
**Presentation:** 2
**Contribution:** 2
**Rating:** 2
**Confidence:** 2

**Summary:**

This paper proposes an approach to generate personalized summaries that captures user's preferences through his past behavior. The method contains 3 steps: predicting the next user behavior from his past behavior sequence; predicting the keyphrases that reflect the user's preference; generating a summary that takes into account the required keyphrases.
Experiments are performed on 2 datasets: PENS and OpenAI-Reddit. The proposed method is shown to outperform the existing methods at large margins.

**Strengths:**

1. The idea of capturing user's preferences through his past behavior, and use them to guide summarization is interesting. (However, the idea needs to be further motivated.)

2. The paper presents quite sophisticated methods to capture user's behavior patterns using RNN. The main idea behind the methods sounds reasonable.

3. The authors attempt to create datasets from the existing ones to perform experiments on personalized summarization. The creation of the datasets represents a significant amount of time. (However, the creation process has some fatal problems)

**Weaknesses:**

1. While personalizing text summaries may be an appealing idea, the paper does not motivate it correctly with relevant application contexts. Especially, the paper argues that the personalization may evolve and depend strongly on the past behaviors of the user, so using a static user profile is not appropriate. It is difficult to see in what application situation such fine-grained personalization is required, and what it may differ from using a static user profile.

2. The main idea of the proposed approach is to predict the next behavior, then the required keyphrases. The keyphrases are then incorporated in text summarization through prompting an LLM. The three tasks are described in the paper as 3 quite independent tasks. It is unclear why user's past behavior may influence the selection of useful keywords. For example, if a user A clicked on a document and another user B did not, why this difference indicate that different summary should be generated for the same test document for these users? It is unclear why the prediction of the next behavior can help determine the best keyphrases for a document.

3. The construction of the datasets is seriously flawed. The goal of the paper is to produce personalized summaries. So, personalized summaries should be used as gold standard. However, no such personalized summarties exist from the initial datasets (PENS and OpenAI-Reddit). The authors hired a set of annotators to create gold standard summaries. These annotators are not the users who have behaved in some way in the past, and may have different interests than the true users. The summaries created by these annotators cannot be considered as personalized summaries. A summary having a high ROUGE or METEOR metric may not be a personalized one. The evaluation based on such datasets do not correspond to the main claim of the paper on personalized summarization.

4. It is mentioned in an appendix (lines 1211-1214) that to deal with the problem of "incomplete representation of the user dynamic preference", summaries (s-nodes) in the test data are incorporated into training data. It is difficult to see the impact of this (because of difficulty to understand the description), but this will clearly create data leak problem.

5. The paper is hard to read. It uses complex notations, and a complex method for user behavior modeling (while may not be necessary). Many of the important details are described in the appendix (e.g. the details on dataset construction). One has to go back and forth to understand all the details. Missing some of those details may lead to wrong interpretations of the results. The tables are difficult to understand because of the lack of proper description. One often has to read the appendix to understand what is meant. Even a simple description of what is intended to show in a table would help a lot.

**Questions:**

see weakness

---

### Note · Authors · 2025-11-12

I have read and agree with the venue's withdrawal policy on behalf of myself and my co-authors.